

# Tropospheric NO₂ retrieval algorithm for geostationary satellite instruments: applications to GEMS

Sora Seo[1], Pieter Valks[1], Ronny Lutz[1], Klaus-Peter Heue[1], Pascal Hedelt[1], Diego Loyola[1], Hanlim Lee[2], and Jhoon Kim[3]

[1]German Aerospace Center (DLR), Remote Sensing Technology Institute (IMF), Oberpfaffenhofen, Germany
[2]Division of Earth Environmental System Science, Major of Spatial Information Engineering, Pukyong National University, Busan, Republic of Korea
[3]Department of Atmospheric Sciences, Yonsei University, Seoul, Republic of Korea

*Correspondence to*: Sora Seo (sora.seo@dlr.de)

**Abstract.** In this study, we develop an advanced retrieval algorithm for tropospheric nitrogen dioxide ($NO_2$) from the geostationary satellite instruments and apply it to Geostationary Environment Monitoring Spectrometer (GEMS) observations. Overall, the algorithm follows previous heritage for the polar orbiting satellites GOME-2 and TROPOMI, but several improvements are implemented to account for specific features of geostationary satellites.

The DLR GEMS $NO_2$ retrieval employs an extended fitting window compared to the current fitting window used in GEMS operational v2.0 $NO_2$ retrieval, which results in improved spectral fit quality and lower uncertainties. For the stratosphere-troposphere separation in GEMS measurements, two methods are developed and evaluated: (1) STRatospheric Estimation Algorithm from Mainz (STREAM) as used in the DLR TROPOMI $NO_2$ retrieval and adapted to GEMS, and (2) estimation of stratospheric $NO_2$ columns from the Copernicus Atmosphere Monitoring Service (CAMS) forecast Cy48R1 model data, which introduce full stratospheric chemistry as it will be used in the operational Sentinel-4 $NO_2$ retrieval. While STREAM provides hourly estimates of stratospheric $NO_2$, it has limitations in describing small-scale variations and exhibits systematic biases near the boundary of the field of view. In this respect, the use of estimated stratospheric $NO_2$ columns from the CAMS forecast model profile demonstrates better applicability by describing not only diurnal variation but also small-scale variations.

For the improved air mass factor (AMF) calculation, sensitivity tests are performed using different input data. In our algorithm, cloud fractions retrieved from the Optical Cloud Recognition Algorithm (OCRA) adapted to GEMS level 1 data are applied instead of GEMS v2.0 cloud fraction. OCRA is used operationally in TROPOMI and Sentinel-4. Compared to GEMS level 2 cloud fraction which is typically set to around 0.1 for clear-sky scenes, OCRA sets cloud fractions close or at 0. The OCRA-based cloud corrections result in increased tropospheric AMFs and decreased tropospheric $NO_2$ vertical columns, leading to better agreement with results from existing TROPOMI observations. The effects of surface albedo on GEMS tropospheric $NO_2$ retrievals are assessed by comparing the GEMS v2.0 background surface reflectance (BSR) and TROPOMI Lambertian-equivalent reflectivity (LER) climatology v2.0 product. The differences between the two surface albedo products and their impact on tropospheric AMF are particularly pronounced over snow/ice scenes during winter. A priori $NO_2$ profiles from the





CAMS forecast model, applied in the DLR GEMS algorithm, effectively capture variations in $NO_2$ concentrations throughout the day with high spatial resolution and advanced chemical mechanism, which demonstrates its suitability for geostationary

satellite measurements.

The retrieved DLR GEMS tropospheric $NO_2$ columns show good capability to capture hotspot signals at the scale of city clusters and describe spatial gradients from city centers to surrounding areas. Diurnal variations of tropospheric $NO_2$ columns over Asia are well described through hourly sampling of GEMS. Evaluation of DLR GEMS tropospheric $NO_2$ columns against TROPOMI v2.4 and GEMS v2.0 operational products show overall good agreement. The uncertainty of DLR GEMS

tropospheric $NO_2$ vertical columns varies based on observation scenarios. In regions with low pollution levels such as open ocean and remote rural areas, retrieval uncertainties typically range from 10 % to 30 %, primarily due to uncertainties in slant columns. For heavily polluted regions, uncertainties in tropospheric $NO_2$ columns are mainly driven by errors in tropospheric AMF calculations. Notably, the total uncertainty in GEMS tropospheric $NO_2$ columns is most significant in winter, particularly over heavily polluted regions with low-level clouds below or near the $NO_2$ peak.

**1 Introduction**

Nitrogen oxides ($NO_x$), the sum of nitrogen dioxide ($NO_2$) and nitrogen oxide (NO), play an important role in many atmospheric chemistry processes in both the stratosphere and troposphere. In the stratosphere, $NO_x$ is involved in photochemical reactions with ozone by acting as a catalyst for ozone depletion while also restraining ozone destruction through the formation of chlorine and bromine reservoir species (Solomon, 1999; Seinfeld and Pandis, 2006). In the troposphere, $NO_x$

serves as essential precursors for ozone formation in the presence of volatile organic compounds (VOCs), influencing the concentrations of hydroxyl radicals (OH) and thereby the lifetime of methane (Sillman et al., 1990). Additionally, $NO_x$ contributes to secondary aerosol formation through gas-to-particle conversion (Shindell et al., 2009). Tropospheric $NO_2$, responsible for both ozone and aerosol production, has significant impacts on air quality, human health, radiative forcing, and global climate change. Given its critical role, monitoring the concentration of $NO_2$ in the atmosphere is important.

Over the past few decades, $NO_2$ column measurements have been provided from polar sun-synchronous low-earth-orbiting (LEO) satellite instruments, including the Global Ozone Monitoring Experiment (GOME), SCanning Imaging Absorption SpectroMeter for Atmospheric CHartographY (SCIAMACHY) (Bovensmann et al., 1999), Ozone Monitoring Instrument (OMI) (Levelt et al., 2006), Global Ozone Monitoring Experiment-2 (GOME-2) (Callies et al., 2000; Munro et al., 2016), and Tropospheric Monitoring Instrument (TROPOMI) (Veefkind et al., 2012). These space-borne remote sensing measurements

have contributed to our understanding of the global distribution of tropospheric $NO_2$ levels, their changes over time and estimates of emissions. However, the LEO instruments only observe $NO_2$ once a day at a specific local time, which has limitations in monitoring of diurnal variations in $NO_2$ due to variations in emissions and chemical reactions throughout the day. Ground-based measurements offer higher temporal sampling of atmospheric compositions within a day, but are limited in terms of spatial coverage. To address the shortcomings of the current atmospheric composition monitoring system, the



Geostationary Air Quality (Geo-AQ) constellation mission, consisting of three geostationary satellite sensors, i.e. Geostationary Environment Monitoring Spectrometer (GEMS) for Asia, Tropospheric Emissions: Monitoring of Pollution (TEMPO) for North America, and Sentinel-4 (S4) for Europe, was coordinated by the Committee on Earth Observation Satellites (CEOS) (Zoogman et al., 2017; Kim et al., 2020).

The GEMS instrument on board the Geostationary Korea Multi-Purpose Satellite-2B (GEO-KOMPSAT-2B), launched in

February 2020, is the first geostationary satellite to monitor air quality at an unprecedented spatial and temporal resolution and has been provide continuous hourly observations over Asia (Kim et al., 2020). GEMS is a step-and-stare UV-visible imaging spectrometer, which combines a scan mirror with a push-broom design. As the mirror scans from east to west, it captures light from a narrow strip on Earth oriented in the north to south direction, reflecting it into the telescope. The spectral coverage of GEMS is 300-500 nm wavelength range with a spectral resolution of 0.6 nm. The imaging process takes 30 minutes, followed

by a transmission time of 30 minutes. The field of view (FOV) of GEMS covers the East and Southeast Asia (5 °S - 45 °N, 75 - 145 °E). The nominal spatial resolution is 7 km x 8 km at a reference location, specifically Seoul, South Korea. However, it is important to note that the spatial sampling distance varies across the geographic coverage area due to projection and curvature effects. $NO_2$ measurements from GEMS can be beneficially used for ground-level concentration estimates and emission strengths. In particular, the hourly sampling with a high spatial resolution allows for a detailed analysis of the diurnal

evolution of $NO_2$ and local distribution of emission sources over Asia, where air quality monitoring has emerged as an important issue due to rapid economic development and urbanization in past decades, for the first time from space.

In this study, we present a $NO_2$ retrieval algorithm designed for geostationary satellites using GEMS measurements. The DLR GEMS $NO_2$ retrieval algorithm is based on a heritage of $NO_2$ retrieval from previous LEO satellites, following a common approach consisting of three key steps: (1) the spectral retrieval of total $NO_2$ slant columns, (2) the separation of slant columns

into stratospheric and tropospheric contributions, and (3) the conversion of tropospheric slant columns to tropospheric vertical columns using air mass factors (AMFs). However, to account for the characteristics of the geostationary satellite, such as hourly sampling, limited geographical coverage, and larger zenith angles, we develop and implement a number of improvements in the DLR GEMS $NO_2$ retrieval algorithm. To estimate the stratospheric contribution and describe the diurnal variation of stratospheric fields, an improved stratosphere-troposphere separation approach using the Copernicus Atmosphere

Monitoring Service (CAMS) global forecast model data is developed and evaluated by comparing it with the results obtained from an existing refined spatial filtering method. For the improved tropospheric AMF calculation, sensitivity tests are performed using different input datasets regarding cloud properties, surface albedo and a priori $NO_2$ profiles. A detailed description of the DLR $NO_2$ retrieval algorithm for the GEMS instrument is provided in Sect. 2. In Sect. 3, we present examples of applying the retrieval algorithm to GEMS measurements, highlighting its capability in monitoring of diurnal variability.

Also, DLR GEMS tropospheric $NO_2$ retrievals are evaluated in comparison with the operational TROPOMI v2.4 and GEMS v2.0 L2 $NO_2$ products. Section 4 contains a summary and conclusions.



## 2. DLR GEMS NO₂ retrieval algorithm

Independent from the operational processing, the scientific NO₂ retrieval algorithm for GEMS instrument is developed at DLR. The DLR GEMS NO₂ retrieval algorithm mainly follows a classical three-step scheme used in the previous satellites, with several improvements to account for specific aspects of GEMS geostationary instrument. First, the retrieval of a total NO₂ slant column density (SCD) from the level 1 radiance and irradiance spectra measured by GEMS using a Differential Optical Absorption Spectroscopy (DOAS) technique (Platt and Stutz, 2008). To determine the tropospheric NO₂ slant column, the stratospheric contribution is estimated and removed from the total slant column, after which both total and tropospheric slant column densities are converted to vertical column densities (VCDs) by the application of AMFs. Each step of the DLR GEMS NO₂ retrieval algorithm is described in detail in the sections below.

### 2.1 DOAS slant column retrieval

The NO₂ slant column is retrieved using the DOAS method (Platt and Stutz, 2008), which quantifies the amount of NO₂ along the atmospheric light path based on Beer-Lambert's law. The main concept of DOAS is to separate the wavelength dependent absorption signal into two components: the high frequency structures of absorption cross sections used for the retrieval of trace gases and the low frequency part arising from scattering by molecules and particles, as well as reflection at the surface, treated as a closure term that is fitted by a low-order polynomial. From the backscattered spectra measured by the instrument, the DOAS retrieval is performed by a least-squares fit from the following equation (1):

$$ln\left[\frac{I(\lambda) + offset(\lambda)}{I^0(\lambda)}\right] = -\sum_g S_g \sigma_g - \alpha_R R(\lambda) - P(\lambda) \qquad (1)$$

The measurement-based term is defined as the natural logarithm of the measured earthshine spectrum $I(\lambda)$ divided by the daily solar irradiance spectrum $I^0(\lambda)$. An intensity offset correction $offset(\lambda)$ is fitted as an additional linear term to correct issues related to incomplete removal of straylight in the spectrometer, inelastic scattering in the ocean water, and dark current in the level 1 spectrum (Paltt and Stutz, 2008; Richter et al., 2011).

In this study, the fitting window for NO₂ is extended from 425 to 480 nm, compared to the fitting window of 432-450 nm currently used in the operational GEMS v2.0 NO₂ retrieval. The use of a large fitting interval generally leads to a reduction in noise on the slant column retrievals. However, significant improvements can be achieved only if the level 1 spectra maintain homogeneous quality over the spectral fitting range. Since saturation effects over bright clouds are detected at shorter wavelengths, and absorptions by NO₂ become relatively less significant improvements with extension to shorter wavelengths below 425 nm (Richter et al., 2011), the lower limit of the fitting window for GEMS NO₂ is determined as 425 nm. In addition, the upper limit is set at 480 nm due to systematic spectral features observed in DOAS fitting at wavelengths above ~ 480 nm. The spectral effect from the absorption of species g is characterised by the fitted slant column density $S_g$ and the associated absorption cross section $\sigma_g$:

- NO₂ absorption at 220 K from Vandaele et al. (2002)





- O$_3$ absorption at 243 K from Serdyuchenko et al. (2014)

- Water vapor (H$_2$O$_{vap}$) absorption at 293 K from Rothman et al. (201), rescaled as in Lampel et al. (2015)

- Oxygen dimer (O$_4$) absorption at 293 K from Thalman and Volkamer (2013)

- Liquied water (H$_2$O$_{liq}$) absorption at 297 K from Pope and Fry (1997), smoothed as in Peters et al. (2014)

- Pseudo cross-section for polarization correction

High-resolution absorption cross-sections are pre-convolved with the GEMS instrument's spectral response function. Given the relative smoothness of these convolved cross-sections, interpolation to the radiance wavelength grid is performed through spline interpolation. The second term on the right-hand side of Eq. (1) is Ring reference spectrum $R(\lambda)$ with the Ring scaling

parameter $\alpha_R$ to describe the filling-in effect of Fraunhofer lines by inelastic rotational Raman scattering, Ring effect. The last term is the closure polynomial approximated by a fourth-order polynomial $P(\lambda)$.

In the NO$_2$ slant column retrieval, a single NO$_2$ cross-section reference spectrum measured at a fixed temperature 243 K is used as a convenient approach. However, as the amplitude of the differential cross-section features shows a temperature dependence, a posteriori temperature correction for the difference between the atmospheric temperature and the reference

cross-section temperature is performed in the air mass factor calculation to account for the temperature sensitivity of NO$_2$ based on Boersma et al. (2004). In addition to the trace gas absorption cross sections, a pseudo cross-section is added in the DOAS fit to correct for remaining polarization correction problems found in the GEMS radiance spectrum.

To assess the uncertainty estimates of GEMS NO$_2$ slant columns, we performed a statistical analysis by quantifying the spatial slant column variability over the pristine tropical Pacific region (Boersma et al., 2007; Zara et al., 2018). This clean reference

region (5 °S – 5 °N and 80 – 130 °E) is divided into small boxes (1° × 1°), ensuring statistically robust sampling and minimizing contributions from other pollutions and geophysical variability. We assume that variability within the box is attributed to random uncertainty originating from noise in the level 1 data and imperfections in the spectral fitting retrieval. It is notable that measurements with relative geometric AMF variability exceeding 5 % are excluded to mitigate the influence of variability in viewing geometry on the results. In practice, slant columns in each box are observed under similar viewing geometries.

Figure 1 shows the distribution of the deviation of NO$_2$ slant columns from the corresponding box mean value for the DLR GEMS and GEMS v2.0 L2 product. The distribution of slant column deviation is a nearly Gaussian shape, and the width (1σ) of the Gaussian function fitted to the distribution can be used as an indicator of the statistical uncertainty estimate. The width of the Gaussian is about $0.90 \times 10^{15}$ and $1.44 \times 10^{15}$ molec cm$^{-2}$ for the DLR GEMS and the GEMS operational v2.0 retrieval, respectively, which suggests a better quality of spectral fitting results in the DLR GEMS, given the use of GEMS L1 v1.2.4

spectra. The improved NO$_2$ slant column retrieval with lower uncertainties in the DLR GEMS is mainly attributed to employing a larger fitting window, which includes more spectral points and provides relevant absorption cross sections. The use of pseudo cross-section for polarization correction has a minor effect on the NO$_2$ fit quality (~0.02 %), leading to a slight improvement in systematic biases.





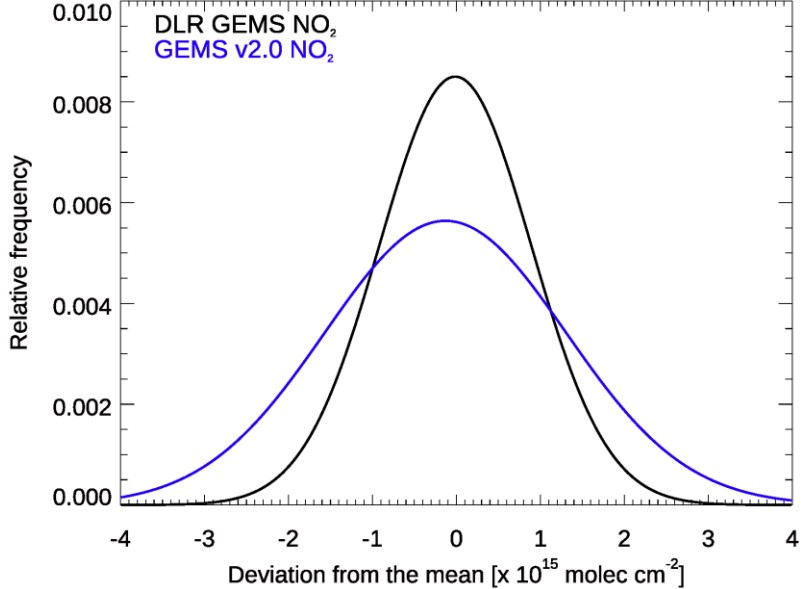

**Figure 1: Gaussian fits for the deviation of GEMS NO$_2$ slant columns from corresponding (1° × 1°) box mean values in the equatorial reference sector (5 °S – 5 °N; 80 – 130 °E) for March 2023 from the DLR GEMS (black) and GEMS operational v2.0 algorithms (blue). The width of the Gaussian provides an estimate of the slant column uncertainty ($\sigma_{DLR\ GEMS} = 0.903 \pm 0.003 \times 10^{15}$ molec cm$^{-2}$, $\sigma_{GEMS\ v2.0} = 1.436 \pm 0.009 \times 10^{15}$ molec cm$^{-2}$).**

## 2.2 Stratosphere-troposphere separation

The slant column retrieved by the DOAS fit contains all NO$_2$ absorption along the atmospheric light path including both the stratosphere and troposphere. To determine the tropospheric NO$_2$ column, which is of primary relevance for air quality, a separation of the stratospheric and tropospheric contributions from the total column is a crucial step. A variety of stratosphere-troposphere separation methods have been developed for global sun-synchronous instruments. However, to be applied in the geostationary satellite instrument, there are following points that need to be resolved in existing stratospheric separation methods. First of all, GEMS samples diurnal variability and requires hourly estimates of stratospheric columns during daytime, which differs from the LEO instruments. The diurnal variability of NO$_2$ in the stratospheric field derived by chemical reactions and dynamics should be accurately described in the geostationary stratosphere-troposphere separation, which demands detailed stratospheric chemistry schemes. Secondly, the number of unpolluted clean pixels in the GEMS FOV for each hour is very limited, which makes it challenging to employ a reference sector method using hourly GEMS measurements. Third, as GEMS measurements cover only a portion of the globe, assimilation approaches where model predictions of stratospheric NO$_2$ columns are adjusted towards the observed satellite NO$_2$ columns have limitations in accounting for the effects of atmospheric transport across the domain boundaries. Therefore, the stratosphere-troposphere separation approaches used in polar orbiting



satellites should be modified and further developed to suit the GEMS geostationary instrument. In this section, we describe and evaluate two different approaches for stratosphere-troposphere separation for GEMS measurements.

### 2.2.1 STREAM adapted to GEMS measurements

The STRatospheric Estimation Algorithm from Mainz (STEAM) is an advanced reference sector method using total $NO_2$

column measurements over clean, remote regions as well as over clouded scenes with negligible tropospheric $NO_2$ abundance (Beirle et al., 2016). STREAM calculates weighting factors for each satellite pixel by assigning a high weight to cloudy observations and a low weight to polluted observations, and the weights are further adjusted in the case of large biases in the tropospheric residues. Based on these weighting factors, stratospheric $NO_2$ fields are derived through a weighted convolution of total columns using convolution kernels, which are designed to be wider at lower latitudes to account for the low zonal

variability assumption of stratospheric $NO_2$ and narrower at higher latitudes to reflect the stronger natural variations. STREAM has been successfully adapted to the polar orbiting satellites and utilized as a complement to the stratosphere-troposphere separation based on data assimilation used in the TROPOMI operational algorithm with the advantage of requiring no model input (Beirle et al., 2016; Song et al., 2021).

While STREAM demonstrates good performances in stratospheric correction for LEO instruments, its applicability to GEMS

observations should be evaluated since geostationary instruments, unlike LEO instruments, have to predict hourly stratospheric fields over a limited geographic domain. Here, we apply the STREAM to GEMS measurements and evaluate its performance. The STREAM settings for GEMS closely resemble those used in TROPOMI measurements, except for the a priori removal of the overall latitude dependency in the reference sector. This adjustment is necessary due to the unavailability of the Pacific reference sector from GEMS measurements. Additionally, we optimized the definition of the spatial convolution kernel to suit

GEMS observation conditions. A pollution weight, representing the potential tropospheric contribution to the total column, is based on our a priori knowledge of the mean spatial distribution of tropospheric $NO_2$ columns derived from three years of TROPOMI measurements (2018-2020). Based on the pollution weight, as well as the cloud weight and tropospheric weight, STREAM estimates stratospheric fields for the hourly scans through a weighted convolution on $0.5° \times 0.5°$ grid pixels.

The adaptation of STREAM to GEMS data successfully yields hourly stratospheric estimates with reasonable values as shown

in Fig. 2. The overall diurnal variability, as well as latitudinal and longitudinal dependencies, are clearly reflected in the stratospheric fields retrieved by STREAM. However, its application to GEMS causes systematic biases near the borders of FOV due to the broader effects of weighted convolution over the limited spatial domain compared to global application. Additionally, STREAM has limitations in estimating small-scale stratospheric features caused by stratospheric dynamics and chemistry, such as free-tropospheric intrusions of $NO_2$, due to relatively wide spatial convolution and a lack of supporting

information.







**Figure 2: Diurnal variation of estimated stratospheric NO₂ columns from STREAM adapted to the DLR GEMS NO₂ retrieval algorithm (described in Sect. 2.2.1) on 10 June 2023. The scan hour spans from 9 June 2023 22:45 UTC (10 June 2023, 07:45 Korea Standard Time) to 10 June 2023 07:45 UTC (10 June 2023, 16:45 KST).**


## 2.2.2 Estimation of stratospheric NO₂ using the CAMS forecast model data

While the application of STREAM to GEMS stratosphere-troposphere separation offers advantages, such as not requiring any inputs from chemical transport models and providing gradients of stratospheric NO₂ within general uncertainties, it is not



accurate enough to capture the diurnal variability of the stratospheric $NO_2$ on small-scales. Therefore, we developed and evaluated an advanced approach for the stratosphere-troposphere separation method for geostationary satellite observations of $NO_2$ columns utilizing the $NO_2$ profile data from the chemical transport model that incorporates comprehensive stratospheric chemistry.

The ECMWF's Integrated Forecast System (IFS) is the global atmospheric model used by the CAMS to provide analysis and
forecast of atmospheric compositions including reactive trace gases, greenhouse gases and aerosols (ECMWF, 2023). Currently, the CAMS global model includes a chemistry scheme based on a CB05-based carbon-bond mechanism with the option to couple with stratospheric chemistry from the Belgian Assimilation System for Chemical ObsErvations (BASCOE) system. The first operational version combining the BASCOE module became available in the CAMS global atmospheric composition forecast system following the upgrade to cycle 48R1 since June 27, 2023 (Chabrillat et al., 2023; Eskes et al.,
2024). This upgrade to CAMS IFS Cy48R1 (referred to as "CAMS forecast" throughout this study) introduced comprehensive stratospheric chemistry based on the BASCOE chemistry scheme, enabling the representation of the diurnal cycle of stratospheric $NO_2$. With a high spatial resolution of $0.4° \times 0.4°$ and 137 vertical layers, the CAMS forecast model is suitable for application in GEMS observations. The current time step of the CAMS forecast output is 3 hours, but this can be improved further in the future. The upgrade of CAMS global forecast model system to Cy48R1 was validated using a number of
independent measurement datasets (Eskes et al., 2023b; Eskes et al., 2024). Validation results demonstrate significant improvements in the ozone profile in the lower-middle stratosphere and stratospheric $NO_2$ due to the inclusion of full stratospheric chemistry. In particular, an evaluation of monthly mean stratospheric $NO_2$ columns from the CAMS forecast model against those retrieved using TROPOMI observations based on the STREAM method showed good agreement in terms of absolute amounts, zonal distributions and temporal variations of stratospheric $NO_2$. The CAMS forecast model well
represents the stratospheric $NO_2$ column amounts within ~10 % deviation range from TROPOMI stratospheric columns and reproduces the general increase in stratospheric $NO_2$ at high latitudes in the summer hemisphere by reflecting the stratospheric chemistry and dynamic (Eskes et al., 2024).

Based on the good validation results of CAMS stratospheric $NO_2$, we estimate the stratospheric $NO_2$ column for GEMS using the CAMS model forecasts of stratospheric $NO_2$ profiles. Although the CAMS global forecast system realistically describes
the variability of stratospheric $NO_2$ fields over time and region, there exists a bias between the model estimates and satellite observations. Hence, it is essential to analyse spatial and temporal bias patterns and apply bias corrections to the model-estimated stratospheric $NO_2$ columns to adapt the model data to the stratosphere-troposphere separation in satellite retrieval algorithms. To calculate the model bias patterns, synthetic initial (geometric) total $NO_2$ columns $V_{init}^{model}$ are first calculated as follows:

$$V_{init}^{model} = \frac{S_{total}^{model}}{M_{strat}} = \frac{V_{total}^{model} \times M_{total}}{M_{strat}} \qquad (2)$$

Modelled $NO_2$ total slant columns $S_{total}^{model}$ are based on the total vertical columns $V_{total}^{model}$ from the CAMS forecast model profile with interpolation to match the GEMS centre pixel coordinate and measurement time. Total AMFs $M_{total}$ and



stratospheric AMFs $M_{strat}$ are derived considering surface properties and cloud information for GEMS orbital data and with CAMS forecast a priori $NO_2$ profiles for the whole atmosphere and between the tropopause height and the top of the atmosphere, respectively. Figure 3 and 4 present the observed initial total $NO_2$ columns from GEMS (a), the simulated initial

total $NO_2$ columns derived from the CAMS forecast profile (b), and differences between the model synthetic and satellite observed initial columns (c) on 10 December 2022, and 10 June 2023, respectively. While both the model simulated and the satellite observed initial columns show similar spatial distributions and value ranges in $NO_2$ columns, biases are apparent, particularly with larger biases over polluted regions, as illustrated in the difference map.

To investigate the patterns of model biases in the stratosphere, it is necessary to focus on pixels where the tropospheric

contribution to the slant column is minimal. Here, we apply a selection rule based on the weighting factor approach used in the STREAM to find pixels with dominant stratospheric contributions (Fig. 3d and 4d). Clouded or clean background pixels, where tropospheric influence is minimal and direct estimates of the stratospheric field are provided, are assigned high weights. Conversely, polluted pixels dominated by the influence of tropospheric sources are assigned low weights close to 0. The weighted model biases are determined by applying the weighting factors to model biases (i.e., the modelled columns minus

the observed columns), which allows for the analysis of the impact of model biases on the stratospheric fields. Assuming that model biases in the stratospheric field depend on latitude and observation time (local time of day), the weighted model biases are fitted with a low-order bivariate polynomial as a function of these variables (see Fig. 5). This polynomial fit for model bias correction acts as a low-pass filter, ensuring zonal smoothness and mitigating artifacts that may arise from unfiltered tropospheric contributions. Model bias patterns are calculated and parameterized on a daily basis, and new bias patterns are

applied accordingly. In Fig. 6, the modelled stratospheric $NO_2$ vertical columns (a, c) and the bias-corrected modelled stratospheric $NO_2$ columns (b, d) are depicted at 04:45 UTC (13:45 KST) on example days in December and June, respectively. Here, the bias-corrected modelled stratospheric $NO_2$ columns are calculated by subtracting the model bias correction terms obtained through parameterization (Fig. 5) from the modelled stratospheric $NO_2$ columns. Therefore, the bias-corrected model stratospheric $NO_2$ columns (Fig. 6b and 6d) used in the stratosphere-troposphere separation in the GEMS $NO_2$ retrieval

algorithm are smaller compared to the original modelled stratospheric columns (Fig. 6a and 6c). Seasonal variability is also evident depending on stratospheric chemistry, with the stratospheric $NO_2$ column lower in winter than in summer at the same scan time.

The stratospheric $NO_2$ column, derived from the CAMS forecast stratospheric model profile as outlined above, exhibits a pronounced diurnal variation (Fig. 7). After sunrise, there is a decrease in the stratospheric $NO_2$ column due to the process of

$NO_2$ photolysis into NO, followed by a quasi-linear increase as sunset approaches (Dirksen et al., 2011). Also, a meridional gradient of stratospheric $NO_2$ is well described depending on the solar zenith angle. While stratospheric fields estimated from the CAMS forecast model approach show an overall similar range and spatial distribution of $NO_2$ columns with those from STREAM (Fig. 2), this method notably captures small-scale stratospheric variations. Given that the stratosphere-troposphere separation method utilizing the CAMS forecast model profile effectively describes the small-scale variations of stratospheric

$NO_2$ over time, which is required for GEMS observations with high spatial and temporal resolution, this approach is selected





as the baseline in the DLR GEMS NO$_2$ retrieval algorithm. It is important to note that CAMS will be the baseline for the Sentinel-4 NO$_2$ retrieval algorithm.

**Figure 3: Maps of the (a) observed initial total NO₂ columns from GEMS, (b) simulated initial total NO₂ columns derived from the CAMS forecast model profile, (c) differences between the model simulated and satellite observed initial columns (b-a in this figure), and (d) weighting factors indicating dominant stratospheric contributions for 10 December 2022 at 04:45 UTC. If the tropospheric NO₂ contribution is significant, the weighting factor (d) is assigned close to 0, and the corresponding pixel is excluded from the sample data for calculating stratospheric model biases.**







Figure 4: Same as Fig. 3, but for 10 June 2023 at 04:45 UTC.





Figure 5: Model bias correction terms parameterized as a function of latitude for each scan time on 10 December 2022 (top) and 10 June 2023 (bottom), respectively.




**Figure 6: Simulated model stratospheric NO₂ vertical columns by CAMS forecast (a, c) and the bias-corrected modelled stratospheric NO₂ vertical columns (b, d) on 10 December 2022 (a, b) and 10 June 2023 (c, d).**



**Figure 7: Diurnal variation of estimated stratospheric NO₂ columns based on the approach utilizing the CAMS forecast model profile data (described in Sect. 2.2.2) on 10 June 2023.**



## 2.3 Air mass factor calculation

The conversion of the slant column into the vertical column is performed by dividing an AMF (M). Given the small optical
depth of NO$_2$, AMF can be derived as

$$M = \frac{\sum_l m_l(i) v_l c_l}{\sum_l v_l} \tag{3}$$

where $m_l$ denotes the box-AMFs in layer l, $v_l$ is the partial column density, and $c_l$ is the correction term for the temperature
dependency of the NO$_2$ cross section (Boersma et al., 2004; Bucsela et al., 2013). $m_l$ depends on retrieval (forward model)
input parameters i, including the GEMS viewing geometry, surface albedo, surface pressure, cloud fraction and cloud pressure.
The box-AMF $m_l$ values are calculated at 452.5 nm, the mid-point wavelength of the spectral fitting window 425-480 nm,
using the radiative transfer model VLIDORT version 2.7 (Spurr, 2006). The box-AMFs are stored in a look-up table (LUT) as
a function of solar zenith angle, viewing zenith angle, relative azimuth angle, surface albedo, surface pressure, and atmospheric
pressure. Pixel-specific box AMFs are obtained by using the best estimates for forward model input parameters and a 6D linear
interpolation. The light path in the troposphere is affected by scattering on air molecules as well as cloud and aerosol particles.
Therefore, the tropospheric AMF calculation should consider the surface albedo, a priori NO$_2$ profiles, and cloud properties.
The impacts of main parameters on the AMF will be discussed in detail in the following sections.

### 2.3.1 Cloud correction

The retrieval of tropospheric NO$_2$ is influenced by the cloud parameters, which derive variations in scene albedo and the photon
path redistribution in the troposphere. In the presence of clouds, to account for cloud-contaminated pixels, the AMF calculation
adopts the independent pixel approximation, which expresses the AMF as a linear combination of a cloudy AMF ($M_{cld}$) and a
clear-sky AMF ($M_{clr}$) (Cahalan et al.,1994; Martin et al., 2002):

$$M = \omega M_{cld} + (1 - \omega) M_{clr} \tag{4}$$

with $\omega$ the radiance weighted cloud fraction derived from the effective cloud fraction ($c_f$):

$$\omega = \frac{c_f I_{cld}}{(1 - c_f) I_{clr} + c_f I_{cld}} \tag{5}$$

where $I_{cld}$ and $I_{clr}$ represent the radiances from the cloudy and clear parts of the pixel, respectively. The values of $I_{cld}$ and $I_{clr}$
depend on GEMS viewing geometries, surface albedo and assumed cloud albedo.

**2.3.1.1 GEMS L2 cloud fraction and cloud pressure**

The operational GEMS cloud algorithm retrieves the effective cloud fraction and cloud centroid pressure based on a LUT that
utilizes the O$_2$-O$_2$ absorption band similar to the OMI cloud retrieval algorithm (Kim et al., 2024). This cloud model assumes
an optically thick Lambertian cloud with a fixed albedo of 0.8. In the first step, the absorption cross-section spectrum of O$_2$-
O$_2$ is fitted in the 460-485 nm range based on the DOAS method yielding the slant column density of O$_2$-O$_2$ $S_{O2-O2}$ together



with the continuum reflectance $R_c$. The $O_3$ cross-section spectrum is included in the spectral fit as it overlaps with the $O_2$-$O_2$

spectrum. In the next step, LUT is used to convert the retrieved quantities $S_{O2-O2}$ and $R_c$ into cloud altitude and effective cloud

fraction. The altitude is transformed into the pressure level with the profile used by the forward model calculations. A detailed

description of the GEMS operational cloud algorithm can be found in Kim et al. (2024). The GEMS L2 cloud product

parameters, specifically the effective cloud fraction and cloud centroid pressure, retrieved by the processer version 2.0 are used

and evaluated for the impacts of cloud correction on GEMS $NO_2$ retrieval.

**2.3.1.2 OCRA cloud fraction adapted to GEMS**

Optical Cloud Recognition Algorithm (OCRA) is the algorithm that provides operational radiometric cloud fraction

information for the European satellite missions dedicated to air quality and trace gas monitoring, including GOME/ERS-2,

SCIAMACHY/Envisat, GOME-2/MetOp-ABC and TROPOMI/S5P (Loyola 1998; Lutz et al., 2016; Loyola et al., 2018) as

well as the NASA EPIC/DSCOVR. OCRA will also be applied to the operational L2 cloud product of the upcoming Sentinel-

4 mission (Loyola et al., 2018). OCRA retrieves the cloud fraction by analysing the colour of the scene using integrated

radiances in UV-vis ranges, separating the sensor measurements into two components: cloud-free background and a remainder

expressing the influence of clouds. Given its flexibility and versatile design, OCRA can be easily adapted and implemented to

other missions and instruments. Here, we apply the OCRA GEMS cloud fraction, which is retrieved by adapting the OCRA

algorithm directly to the GEMS level 1 radiance and irradiance data, as a complement of the operational GEMS cloud fraction.

The adaptation of OCRA to the GEMS L1 data includes the following components. For the clear-sky background, we use

composite maps generated with the EPIC/DSCOVR instrument. See Molina Garcia (2022) for details about the generation of

those maps. OCRA also includes a pre-processing step which intends to minimize cloud fraction overestimation at extreme

viewing geometries. An empirical correction scheme for the GEMS L1 reflectance windows used by OCRA is computed based

on four full months of GEMS L1 data to cover all the seasons. The correction for each OCRA fitting window is then based on

a polynomial fit to monthly mean reflectance data as a function of the viewing zenith angle. The final calculation of the OCRA

GEMS radiometric cloud fraction is then performed based on the viewing angle dependency corrected reflectance. Sensitivity

tests of GEMS tropospheric $NO_2$ vertical column retrieval depending on different cloud fractions of each product will be

described in the following section 2.3.1.3.

**2.3.1.3 Evaluation of cloud corrections**

In this section, we evaluate the influence of cloud correction on the tropospheric $NO_2$ column retrieval using two different

cloud fraction datasets described in Sect. 2.3.1.1 and 2.3.1.2. Figure 8 compares the cloud fraction from the OCRA applied to

GEMS L1 data and the GEMS L2 cloud v2.0 product for the same scan time on the same day. Overall, there is good agreement

in both the cloud structures and value ranges between the two datasets. The correlation between the GEMS L2 effective cloud

fraction and the OCRA cloud fraction is approximately 0.97. However, there are clear spatial patterns that the OCRA cloud

fraction is smaller by about 0.1 for clear-sky scenes, whereas larger for fully cloudy scenes compared to the GEMS L2 cloud





fraction as identified in the difference map (Fig. 8c). Also, regions with extreme observation geometries, i.e. with large solar

zenith angle and viewing zenith angle, differences between the OCRA and GEMS cloud fraction is detected. At zenith angles roughly above 60°, the GEMS cloud fraction shows a constant background signal which is less pronounced in the OCRA cloud fraction due to the application of viewing zenith angle correction.

Figure 9 shows the tropospheric $NO_2$ columns retrieved using both the OCRA and GEMS L2 cloud fraction for the same scan time, demonstrating the impact of cloud corrections with different cloud products. In many cases of cloudy scenes, the cloud

top is generally above the $NO_2$ pollution present in the boundary layer, and the enhanced tropospheric $NO_2$ columns cannot be detected by satellite instruments if the clouds are optically thick. Therefore, tropospheric $NO_2$ columns are displayed for GEMS observations with the cloud fraction less than 0.4. It should be noted that the altitude-dependent AMF for a partly cloudy pixel implicitly includes a correction for the $NO_2$ columns lying below the cloud, referred to as ghost columns, via the cloudy-sky AMF $M_{cld}$ (where $m_l = 0$ for layers below the cloud top pressure). As indicated in the difference map (Fig. 9c), the use of

OCRA cloud fraction results in a reduction in tropospheric $NO_2$ columns, particularly in polluted regions, by up to $2.0 \times 10^{15}$ molec cm$^{-2}$. To examine the case in more detail, we zoomed in and displayed the cloud fraction and tropospheric $NO_2$ columns, including TROPOMI products that overpassed the study domain within one hour of the GEMS scan (Fig. 10). The cross-comparison of cloud fractions using the TROPOMI operational product revealed that OCRA cloud fraction values are closer to TROPOMI cloud fraction in clear-sky scenes, showing better agreement compared to GEMS L2 cloud fraction. Although

the GEMS L2 cloud fraction shows small values around 0.1, these scenes seem to be identified as cloud-free (i.e. cloud fraction is set to 0) for almost clear-sky conditions with the retrieved cloud height very close to the surface height in both the OCRA and TROPOMI cloud products.

Figure 11 shows an example of the box-AMFs derived using the OCRA and GEMS L2 cloud fraction for clear-sky scenes over Shanghai (32.22 °N, 121.37 °E) and Qingdao (35.05 °N, 119.57 °E) in eastern China on March 14, 2023. The cloud

fractions and the calculated tropospheric AMFs are also indicated. Compared to the cloud correction using the GEMS operational cloud fraction, employing OCRA cloud correction increases the tropospheric AMF by ~28 %, while concurrently reducing the tropospheric $NO_2$ column by ~22 %. Notably, under clear-sky conditions, the GEMS L2 cloud fraction of 0.1 makes the retrieval less sensitive to $NO_2$ below the cloud than the OCRA-based cloud correction, where the cloud fraction is at or close to 0. This effect is particularly pronounced in polluted urban areas where elevated levels of $NO_2$ are located in the

boundary layer, resulting in larger differences in retrieved tropospheric $NO_2$ columns due to cloud corrections with different cloud fraction values.



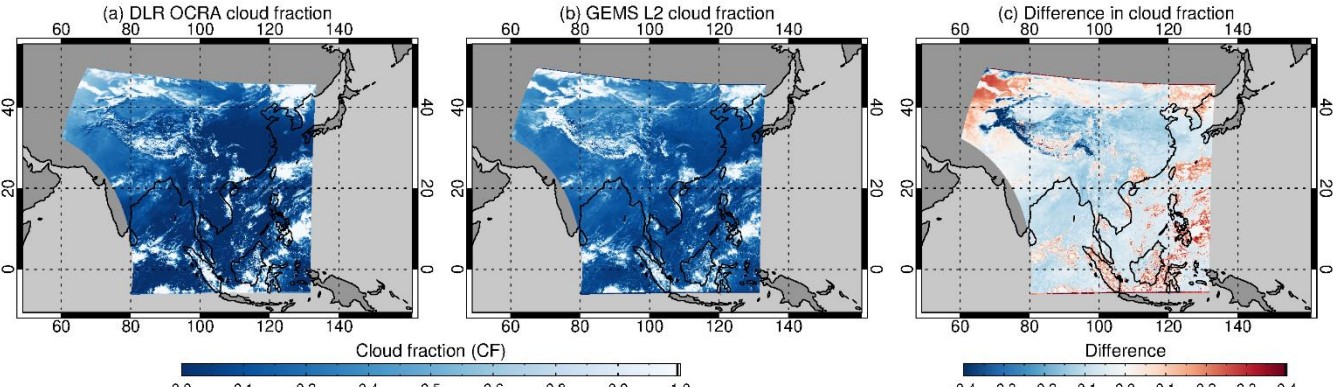

**Figure 8: Maps of cloud fractions from (a) OCRA adapted to GEMS level 1 data, and (b) GEMS L2 v2.0 cloud product for 14 March**
**2023 at 04:45 UTC. (c) Differences in cloud fractions (OCRA – GEMS L2 v2.0; a – b in this figure) for the corresponding scene.**

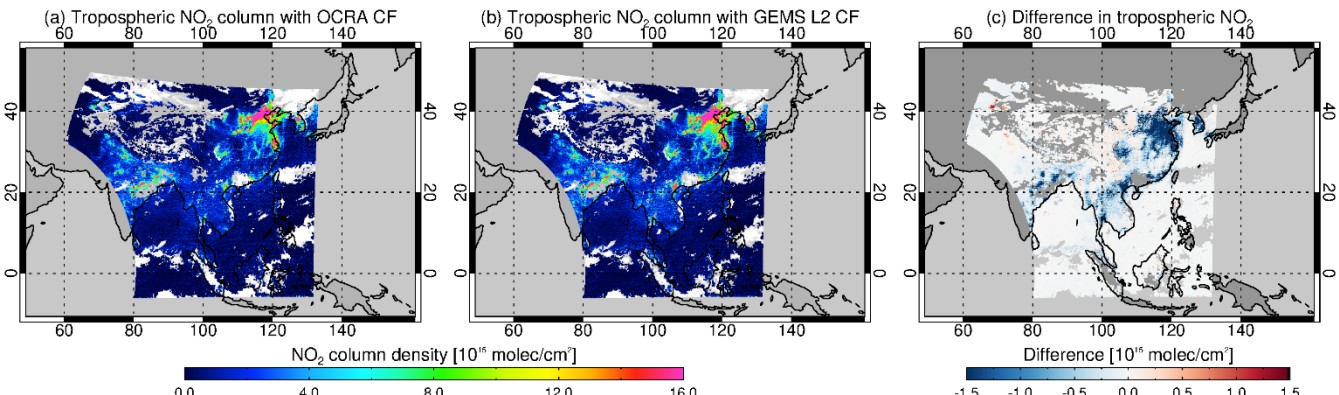

**Figure 9: Tropospheric NO₂ columns retrieved using (a) OCRA cloud fractions (displayed in Fig. 8a), and (b) GEMS L2 v2.0 cloud fractions (displayed in Fig. 8b) for 14 March 2023 at 04:45 UTC. (c) Differences in tropospheric NO₂ columns applying the OCRA-**
**based cloud corrections and GEMS L2 O₂-O₂ based cloud corrections (a – b in this figure). Only measurements with cloud fraction ≤ 0.4 are included.**





**Figure 10: Zoomed maps for cloud fractions from (a) OCRA adapted to GEMS level 1 data, (b) GEMS L2 v2.0 cloud product, and (c) TROPOMI L2 v2.4 cloud product over polluted areas including eastern China and Korea peninsula for 14 March 2023 at 04:45 UTC. GEMS tropospheric NO₂ vertical columns retrieved using (d) OCRA cloud fractions (displayed in Fig. 10a) and (e) GEMS L2 v2.0 cloud fractions (displayed in Fig. 10b). (f) TROPOMI L2 v2.5 tropospheric NO₂ columns for the corresponding scene.**




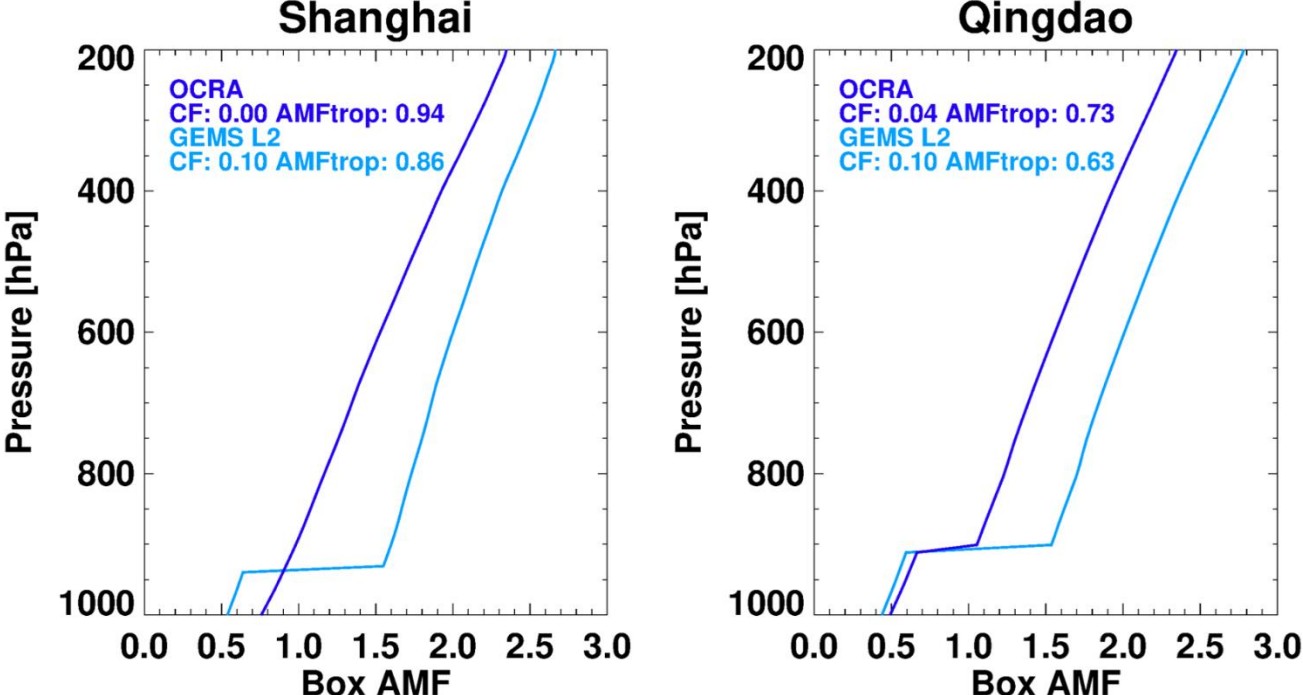

**Figure 11: Comparisons of box-AMFs derived using the OCRA and GEMS L2 cloud fraction for clear-sky scenes over Shanghai (32.22 °N, 121.37 °E) and Qingdao (35.05 °N, 119.57 °E) in eastern China on 14 March 2023 at 04:45 UTC. Cloud fractions and calculated tropospheric AMFs for each case are denoted.**

### 2.3.2 Surface albedo

Surface albedo plays an important role in the accurate retrieval of trace gas columns and cloud properties (Govaerts et al., 2004). Previous studies accessing the accuracy of $NO_2$ columns from satellite measurements show that errors in $NO_2$ retrievals, particularly over major continental source regions, are dominantly affected by errors in the air mass factors caused by imprecise estimates of the surface reflectance (Boersma et al., 2004; Kleioppl et al., 2008). Given the significant impact of surface albedo on the sensitivity of backscattered radiance to boundary layer $NO_2$, even a minor absolute error of 0.01 in surface reflectance can potentially introduce approximately a 15 % error in tropospheric $NO_2$ columns in polluted regions (Boersma et al., 2004). In many retrieval algorithms for trace gases from space-borne spectrometers, the surface reflectivity is described as Lambertian-equivalent reflectivity (LER), assuming isotropic surface reflection. While this simplified approach may be justified when viewing and solar illumination angles are constrained to a narrow range, it disregards the directional dependence of surface reflectivity, which can be described by a bidirectional reflectance distribution function (BRDF) (Tilstra et al., 2017). Radiative transfer models are able to include the BRDF for every order of reflection by the model surface to account for the dependence of surface reflectivity on illumination and observation direction. However, implementing the full BRDF is



computationally intensive in many practical situations, and certain radiative transfer codes may lack proper BRDF handling capabilities. Given these challenges, most operational satellite-based trace gas retrieval algorithms have used LER climatology databases, typically constructed on a gridded monthly basis using statistical values derived from multiple years of satellite

observations. Recently, to complement the traditional LER climatology, several new databases which contain a directional dependence of the albedo values have developed, such as the geometry-dependent effective Lambertian-equivalent reflectivity (GE_LER) (Loyola et al., 2020), the directionally dependent Lambertian-equivalent reflectivity (DLER) (Tilstra et al., 2023). The effect on the angular dependence of surface reflectivity should be considered more carefully for the geostationary satellite due to its hourly sampling, relatively large viewing and solar zenith angles, and high spatial resolution (Govaerts et al. 2004).

To address the atmospheric scattering effects and the anisotropy of the surface reflectance throughout the day, the GEMS operational surface reflectance algorithm operates in two modes, online and offline, consisting of three main steps: (1) atmospheric correction, (2) bi-directional reflectance distribution function (BRDF) modelling, and (3) background surface reflectance (BSR) retrieval. The atmospheric correction process involves computing the top-of-atmosphere (TOA) radiance and converting it to top-of-canopy (TOC) reflectance using the Second Simulation of the Satellite Signal in the Solar Spectrum

Vector (6SV) radiative transfer model (Vermote et al., 2006). Next, the Roujean BRDF model (Roujean et al., 1992), one of the semi-empirical BRDF models, is employed for BRDF modelling to characterize the anisotropic properties of land surfaces. Finally, the GEMS BSR is derived using the BRDF parameters obtained from the land surface area on the preceding day. In the marine area, where the same assumptions as those for land do not apply, the minimum of Rayleigh-corrected reflectance within a 15 days period is utilized. The BRDF retrieval is performed offline and is applied in the online BSR retrieval. A

detailed description of the GEMS BSR algorithm is provided by Sim et al. (2024).

Currently, the GEMS BSR v2.0 product is used as input data for GEMS operational aerosol, cloud, and trace gas retrievals. In this study, we compare the GEMS BSR with the TROPOMI LER climatology v2.0 interpolated linearly in time and distance from the GEMS pixel, and evaluate the impact of surface reflectivity on GEMS tropospheric $NO_2$ retrievals. Figure 12 and 14 show the GEMS BSR at 448 nm, TROPOMI surface LER at 463 nm, and their differences at 04:45 UTC on 9 December 2022

and 10 June 2023, respectively. Overall, the GEMS BSR value is higher than the TROPOMI LER value over land by about 0.1, while it is lower by about 0.05 over the sea. GEMS BSR shows larger differences in surface albedo between the land and ocean, with a distinct boundary along the coastline compared to TROPOMI surface LER. In summer, the spatial pattern remains generally consistent, with GEMS BSR showing higher values over land and lower values over water. However, in winter, the spatial pattern of differences between the two databases is relatively inconsistent and varies depending on the region

and observation period. This inconsistency may be attributed that TROPOMI LER climatology may differ from the actual surface conditions, especially in snow/ice scenes. Additionally, systematic biases in GEMS BSR are more pronounced in winter due to gap-filling close to 0 in cases where no data are available, particularly under high viewing zenith angles and solar zenith angle conditions, such as in Mongolia, northern Tibet and Xinjiang.

The difference in tropospheric $NO_2$ columns depending on surface albedo products is larger in winter ($2.5\times10^{14}$ molec cm$^{-2}$ on average) compared to summer ($1.5\times10^{14}$ molec cm$^{-2}$ on average) (Fig. 13 and 15). Moreover, the larger impact of surface



albedo is observed over polluted regions, such as in eastern China and northern India. Since the surface BRDF increases with larger zenith angles toward the northwest of the GEMS domain, the differences in surface albedo and retrieved tropospheric NO$_2$ columns may increase accordingly. However, the northwestern part of the domain, including Mongolia, Xinjiang and Tibet, predominantly consists of clean background regions, which results in relatively small differences in tropospheric NO$_2$

column values themselves. In this study, we use the GEMS BSR v2.0, which is provided for each scan hour on a daily basis, in the DLR GEMS NO$_2$ retrieval algorithm to account for BRDF effects. However, current remaining issues in GEMS BSR v2.0 such as land-sea discontinuity and uncertainties in TOA reflectance attributed to GEMS level 1 spectra might cause errors in GEMS AMF calculation. These issues will be resolved using the improved GEMS BSR v3.0 product which will be released near future.


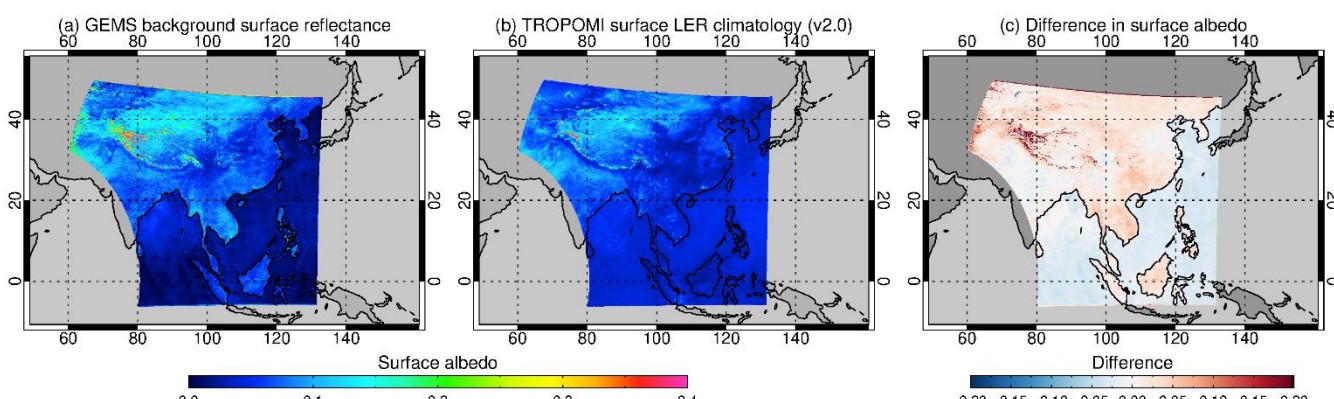

**Figure 12: Maps of surface albedo from (a) GEMS BSR v2.0 product, and (b) TROPOMI surface LER climatology v2.0 product spatiotemporally matched on GEMS pixels for 10 June 2023 at 04:45 UTC. (c) Differences in surface albedo (GEMS BSR v2.0 – TROPOMI surface LER climatology v2.0; a – b in this figure) for the corresponding scene.**


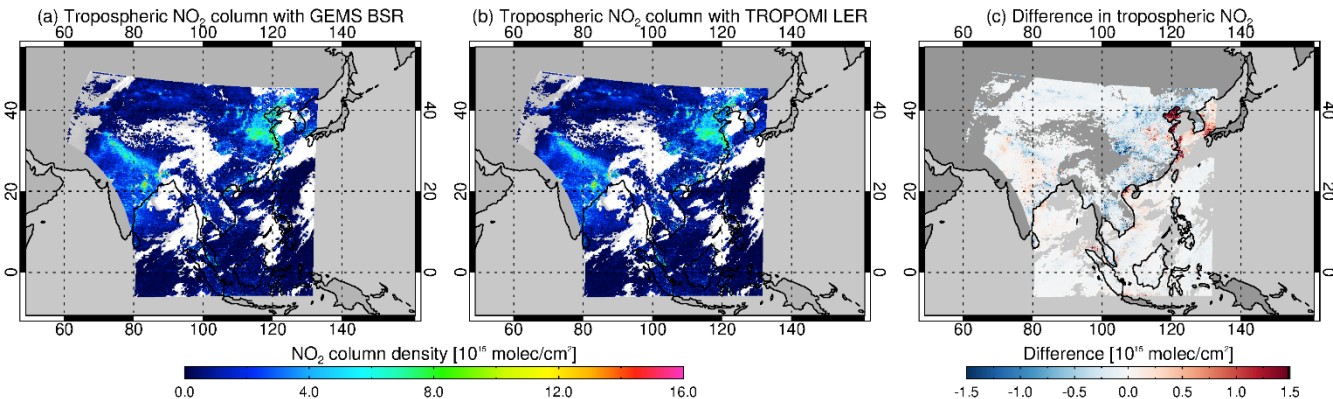

**Figure 13: Tropospheric NO$_2$ columns retrieved using (a) GEMS BSR v2.0 product (displayed in Fig. 12a), and (b) TROPOMI surface LER climatology v2.0 (displayed in Fig. 12b) for 10 June 2023 at 04:45 UTC. (c) Differences in tropospheric NO$_2$ columns applying the GEMS BSR v2.0 and TROPOMI surface LER climatology v2.0 (a – b in this figure). Only measurements with cloud**
**fraction ≤ 0.4 are included.**



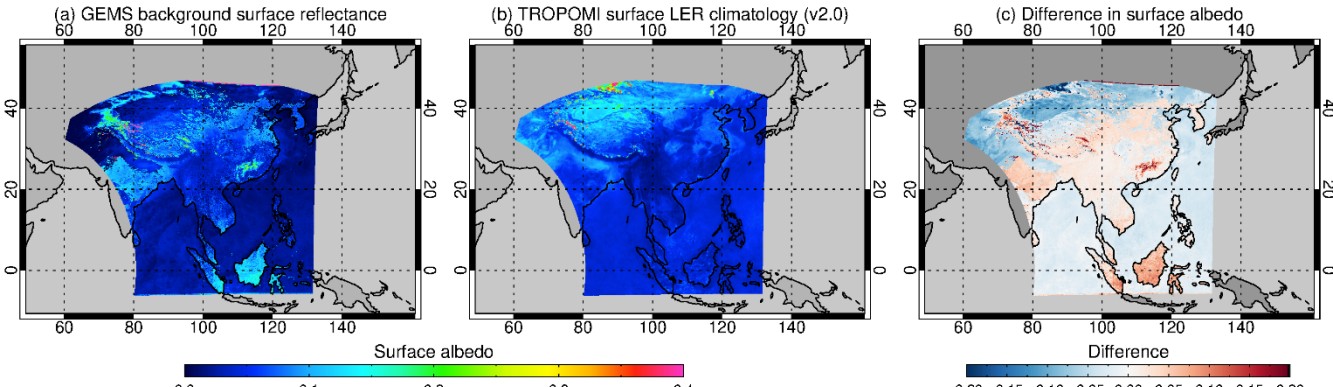

**Figure 14: Same as Fig. 12, but for 9 December 2022 at 04:45 UTC.**

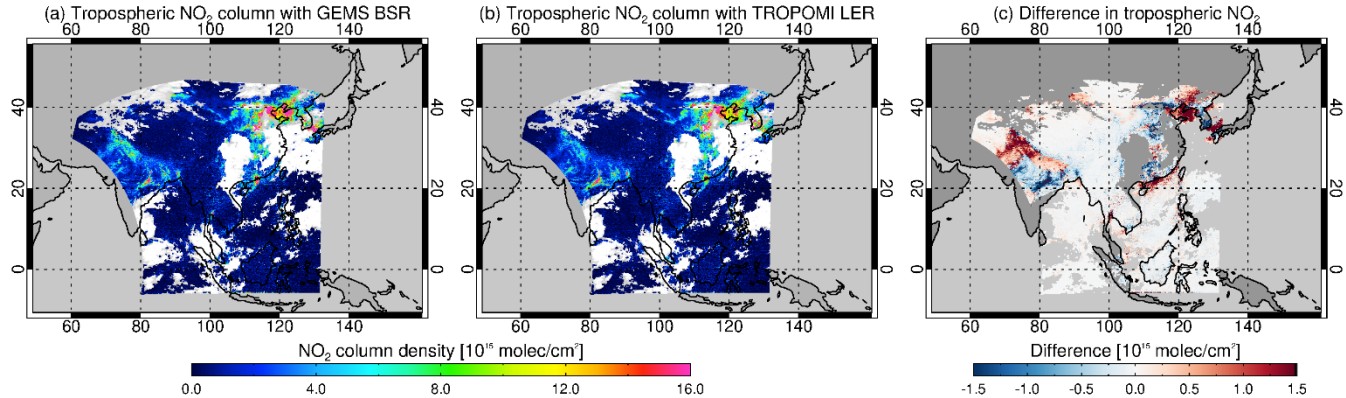


**Figure 15: Same as Fig. 13, but for 9 December 2022 at 04:45 UTC.**

### 2.3.3 A priori NO₂ profile

To account for the varying sensitivity of $NO_2$ at different altitudes, a vertical profile shape of $NO_2$ should be considered in the

AMF calculation. The chemistry transport model is regarded as the optimal source for an a priori $NO_2$ profile. In this study, the CAMS forecast model from cycle 48r1, used in the stratosphere-troposphere separation, is also applied in the tropospheric AMF calculation, which is beneficial to maintain consistency in the GEMS $NO_2$ retrieval. The a priori profile values are computed at the center of GEMS ground pixel through linear interpolation based on distances to neighbouring CAMS forecast grid centers. Additionally, the model profile values are linearly interpolated with respect to the satellite observation time.

Compared to the TM5-MP a priori profile with a 1° x 1° spatial resolution used in TROPOMI $NO_2$ retrieval, the improved spatial resolution as well as more up-to-date chemistry and emissions for trace gas species in the CAMS forecast model enhance





the capability to capture local $NO_2$ distributions, especially in regions with large heterogeneity and variability (Eskes et al., 2024).

Figure 16 shows the diurnal variation in the CAMS forecast a priori $NO_2$ profiles over Seoul, South Korea (37.53° N, 120.01
°E), for summer (15 June 2023) and winter (12 December 2022), respectively. Both selected days are weekdays. The CAMS forecast a priori profiles describe the diurnal evolution of the $NO_2$ volume mixing ratios. In the morning, typically commuting hours from 00:45 to 01:45 UTC (09:45 to 10:45 KST), source emissions increase with the elevated traffic, leading to the highest surface $NO_2$ concentration with a shallow boundary layer. As noon approaches, $NO_2$ concentrations decrease due to diminished source emissions (lower traffic flow) compared to the morning, as well as stronger vertical mixing facilitated by
increased temperature and boundary layer heights. Also, the increased temperature and solar radiation intensity during midday enhance photochemical reactions, which consequently increase the chemical loss of $NO_2$ (An et al., 2016). Tropospheric $NO_2$ columns, which show a minimum during midday, begin to increase again with increased traffic volume near the end of work hours, from 06:45 to 07:45 UTC (15:45 to 16:45 KST), a pattern well described in the CAMS forecast a priori $NO_2$ profile.

In addition to the diurnal variation, the vertical distribution of CAMS $NO_2$ varies according to the season. In winter, high $NO_2$
concentrations are distributed close to the surface level due to abundant $NO_x$ emissions from sources such as traffic and industry, trapped within a shallow boundary layer. On the other hand, in summer, the higher temperature and elevated boundary layer promote stronger convective activity and increased dilution, while stronger solar radiation facilitates the photolysis of $NO_x$. One thing to note is that CAMS forecast $NO_2$ volume mixing ratios and vertical distributions may vary depending on various factors, including meteorological conditions (temperature, wind speed and direction, relative humidity, precipitation,
and radiation) as well as emission patterns on weekdays versus weekends.

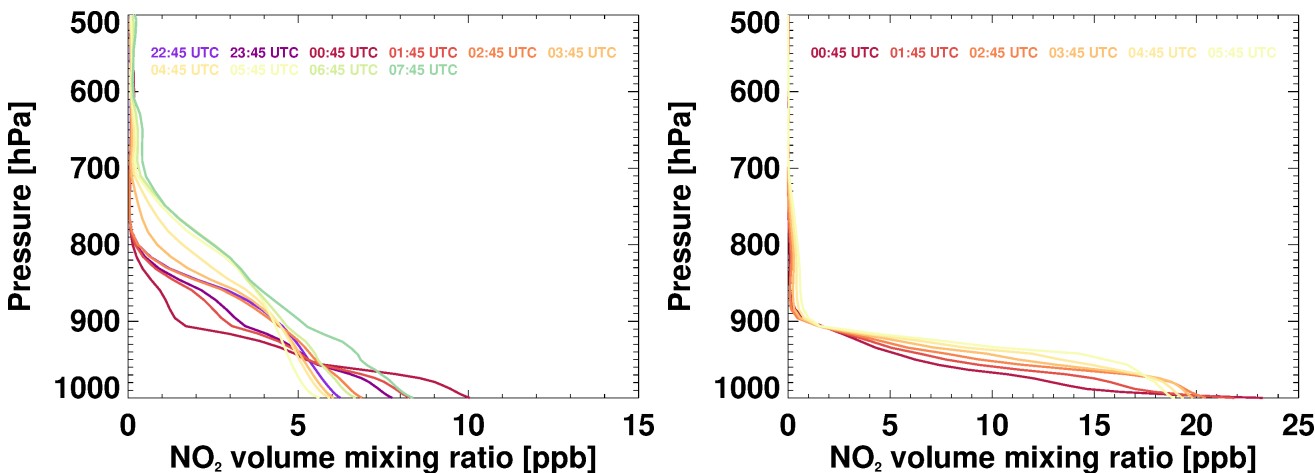

**Figure 16: A priori $NO_2$ profiles from the CAMS forecast model for GEMS each scan hour over the Seoul metropolitan area (37.53° N, 120.01 °E) on 15 June 2023 (left) and 12 December 2022 (right), respectively.**






## 3. DLR GEMS tropospheric NO₂ retrieval evaluation

### 3.1 Examples of GEMS tropospheric NO₂ columns

Figure 17 shows the monthly mean DLR GEMS tropospheric NO$_2$ vertical columns for each scan time from 22:45 to 07:45 UTC (07:45 to 16:45 KST) on a 0.1° x 0.1° grid in June 2023. Only measurements with cloud fractions less than 0.25 are
included. High tropospheric NO$_2$ columns are evident over urban and industrial regions in eastern China, South Korea and northern India. Hotspot signals are also detected as city cluster scale, such as Beijing, Tianjin, Hebei, and Shanghai located in north-northeast China, Chongqing and Chengdu in southwest China, Shenzhen and Guangzhou in south China, and Seoul in South Korea, Tokyo and Osaka in Japan, Delhi and Kolkata in India. In these megacities and surrounding areas, characterized by high population density and anthropogenic sources including motor vehicles, power plants and industries, diurnal variations
in tropospheric NO$_2$ are clearly monitored by the hourly geostationary satellite observations.

Due to its relatively short chemical lifetime with several hours in the boundary layer during daytime, the spatial and temporal variability of tropospheric NO$_2$ is strongly affected by its emissions from sources and meteorological factors such as wind speed, temperature, humidity and illumination (Beirle et al., 2003). The highest levels of tropospheric NO$_2$ are mainly observed in the morning, primarily due to abundant emissions of NO$_x$ from commuter traffic. In urban regions located to the east of the
GEMS domain, such as Tokyo, Seoul and cities in eastern China, the highest tropospheric NO$_2$ columns are found between 23:45 and 01:45 UTC, corresponding to their local morning time, while in India located in the west of the GEMS domain, peak levels are observed between 02:45 and 03:45 UTC due to time zone differences. During the noontime, given sufficient ultraviolet radiation, NO$_2$ is photolyzed to produce NO and oxygen atoms, resulting in a decrease in tropospheric NO$_2$ levels. However, the tropospheric NO$_2$ level begins to increase again in the later afternoon (07:45 UTC in the eastern part of the
GEMS domain), approaching commuting times due to increased traffic in cities. Overall, GEMS tropospheric NO$_2$ retrievals over urban regions exhibit the highest peak in the local morning hours, decreasing to a minimum value around noon, and then increasing again in the afternoon. This pattern is consistent with previous studies that monitored the diurnal variation of tropospheric NO$_2$ using ground-based measurements, demonstrating that the temporal variability of NO$_2$ driven by source emissions and photochemistry over a large coverage can be effectively monitored by the geostationary satellite instrument
(Zhao et al., 2016; Li et al., 2021). Also, spatial gradients of NO$_2$ from city centers to surrounding areas are detected from GEMS observations. Tropospheric NO$_2$ columns, which are high within the megacities due to heavy anthropogenic pollution generated by emissions from a high number of vehicles and industries, decrease with distance as a result of the diminishing NO$_x$ sources and horizontal transport and smearing (Beirle et al., 2011).

Tropospheric NO$_2$ columns are significantly low over rural and open ocean areas. Although elevated levels of tropospheric
NO$_2$ are detected along ship tracks, such as those starting from Singapore, and coastlines near ports and industries, most ocean and rural regions exhibit background levels of tropospheric NO$_2$ due to low anthropogenic emissions. In these pristine areas, NO$_2$ sources are primarily from natural emissions, including production by lightning and microbiological processes in soils (Weng et al., 2020; Li et al., 2021). Hence, the diurnal variation of tropospheric NO$_2$ in rural areas differs from that in polluted





urban areas, with a gradual increase observed from morning to noon. This gradual increase is attributed to biogenic $NO_x$

emissions, which are affected by vegetation, temperature, moisture and radiation (Weng et al., 2020).

**Figure 17: Diurnal variations of the monthly averaged DLR GEMS tropospheric NO₂ columns at each scan hour on 0.1°× 0.1° grid in June 2023. Only measurements with cloud fractions ≤ 0.25 are used in the creation of monthly mean gridded data.**





## 3.2 Comparison with TROPOMI v2.4 NO$_2$ product

To evaluate the DLR GEMS tropospheric NO$_2$ columns, comparisons with independent TROPOMI v2.4 tropospheric NO$_2$ columns are performed. While GEMS observes the domain covering East and Southeast Asia on an hourly basis, ranging from 6 observations in winter to 10 in summer per day, TROPOMI measures the area sequentially per orbit. TROPOMI measurements covering the eastern part of the GEMS FOV, including Japan, start at approximately 03:00 UTC, with the last measurements over the western part of the GEMS FOV, including India, occurring around 08:00 UTC. Therefore, to compensate for the different temporal compatibility between the two satellites, the GEMS data scanned within ± 30 minutes of the TROPOMI observation time is matched. Cloud screening is applied based on the cloud radiance fraction on both products, with a cloud radiance fraction < 0.5. These spatiotemporally matched DLR GEMS and TROPOMI NO$_2$ data are regridded at a resolution of 0.1° × 0.1° to create a comparable dataset. The daily gridded DLR GEMS and TROPOMI tropospheric NO$_2$ vertical columns are compared for June 14, 2023 and December 14, 2022, respectively (Fig. 18 and 19). Figure 20 displays scatter plots for tropospheric NO$_2$ vertical columns between the two datasets on these days.

Overall, the DLR GEMS tropospheric NO$_2$ columns show good agreement with TROPOMI v2.4 tropospheric NO$_2$ columns with correlation coefficients of 0.87 and 0.96, and linear regression slopes of 1.01 and 1.20, respectively (Fig. 20). The datasets also agree well in terms of spatial distribution and value range of tropospheric NO$_2$. However, systematic biases are evident as shown in the difference map (Fig. 18c and 19c). Notably, the GEMS tropospheric NO$_2$ columns tend to have lower values in the open ocean, particularly in the southern part of GEMS FOV, while exhibiting higher values with positive biases over land, especially in polluted regions or the easternmost and westernmost of the domain. This systematic spatial pattern is mainly consistent regardless of season. The observed systematic biases in tropospheric NO$_2$ columns between the DLR GEMS and TROPOMI are mainly attributed to differences in slant columns and tropospheric AMFs calculated using different input datasets.

Considering the different viewing geometries between GEMS and TROPOMI, when comparing the initial NO$_2$ vertical columns scaled by the geometric AMFs to the slant columns, GEMS initial (geometric) vertical columns show a systematically larger gradient in the north-south direction than TROPOMI initial columns. Specifically, the GEMS initial columns indicate lower values with a negative bias in the southern GEMS FOV, while showing higher values with a positive bias toward the northern GEMS FOV compared to TROPOMI initial columns. This phenomenon is also noted in Zhang et al. (2023). The pronounced systematic gradient in the north-south direction in GEMS NO$_2$ slant column retrievals is primarily attributed to the GEMS L1 v1.2.4 spectra. We found systematic variations in the mean spectral fitting residuals along the swath from the north to south direction (not presented here), indicating a need for improvements in the GEMS level 1 data, including advanced radiometric calibration and residual correction. In the DLR GEMS tropospheric NO$_2$ retrieval, the systematic biases along the north-south direction are partially compensated by applying the model bias correction in the stratosphere-troposphere separation (refer to Fig. 5 in Sect. 2.2.2). It is expected that better consistency with NO$_2$ retrievals from TROPOMI will be





achieved once the remaining systematic biases initially originating from the $NO_2$ slant columns are resolved through the application of improved GEMS level 1 spectra.

In addition to the systematic bias in GEMS $NO_2$ slant columns, tropospheric AMF calculations significantly influence the difference in tropospheric $NO_2$ vertical columns between GEMS and TROPOMI. Not only the viewing geometries differ, but the input datasets used to calculate the tropospheric AMF also differ between the two instruments. As described in Sect. 2.3, the DLR GEMS $NO_2$ retrieval algorithm employs the DLR OCRA cloud fraction, GEMS cloud centroid pressure v2.0, GEMS BSR v2.0, and CAMS forecast $NO_2$ profile for the AMF calculations. On the other hand, the TROPOMI $NO_2$ processor v2.4

uses the FRESCO-S cloud product (cloud fraction retrieved from the $NO_2$ spectral window at 440 nm and cloud pressure from the FRESCO-wide utilizing the NIR spectral range), TROPOMI DLER v2.0, and TM5-MP $NO_2$ profile for the AMF calculations. These different ancillary data inputs used for the AMF computation between DLR GEMS and TROPOMI may lead to significant discrepancies in retrieved tropospheric $NO_2$ vertical columns. In particular, differences in tropospheric $NO_2$ columns due to the influence of tropospheric AMFs are pronounced over polluted regions such as eastern China, South Korea,

and northern India. While tropospheric AMF calculations are influenced by all ancillary data (surface albedo, terrain height, cloud parameters, and trace gas profile), one of the key factors contributing to the differences in tropospheric $NO_2$ columns between DLR GEMS and TROPOMI is a priori $NO_2$ profiles. The positive biases observed in GEMS tropospheric $NO_2$ columns over eastern China during winter are largely influenced by the CAMS forecast a priori $NO_2$ vertical profiles. The CAMS forecast model profiles used for GEMS have significantly higher surface layer $NO_2$ concentrations in these polluted

regions compared to the TM5-MP profiles used for TROPOMI, consequently resulting in lower tropospheric AMFs and higher tropospheric $NO_2$ vertical columns.

Furthermore, there may be additional factors contributing to differences in tropospheric $NO_2$ vertical columns between the DLR GEMS and TROPOMI v2.4 product. For example, when thick aerosol layers occur over eastern China in spring, the GEMS operational algorithm which uses the visible $O_2$-$O_2$ band often fails to detect these layers and estimates the cloud

pressure close or equal to the surface pressure. In contrast, TROPOMI cloud pressure retrieval from FRESCO-wide using the NIR O2-A band has more sensitivity and yields more realistic elevated height (lower cloud pressure). In such cases (elevated aerosol layers are present), GEMS tropospheric $NO_2$ vertical columns tend to be lower than those from TROPOMI due to underestimated cloud/aerosol levels in terms of altitude.



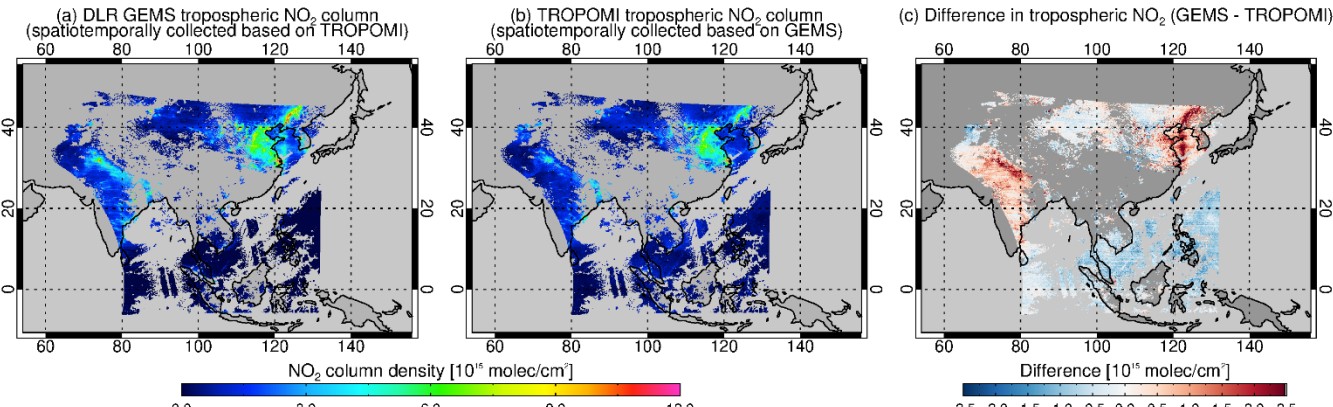

**Figure 18: Comparison between DLR GEMS and TROPOMI operational v2.4 tropospheric NO₂ columns on 14 June 2023. Both tropospheric NO₂ columns are regridded to a 0.1° × 0.1° resolution based on temporally and spatially collocated data for the corresponding date.**

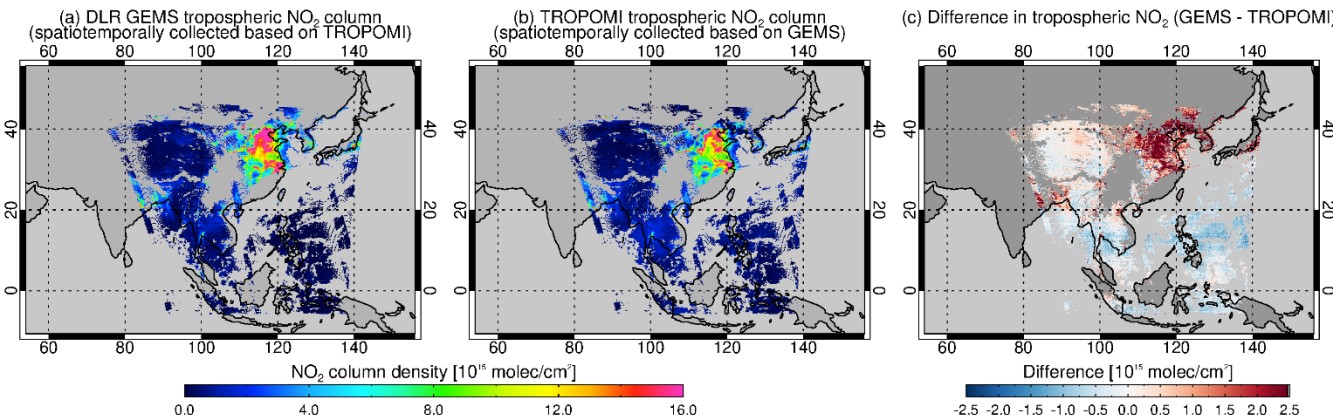

**Figure 19: Same as Fig. 18, but for 14 December 2022.**





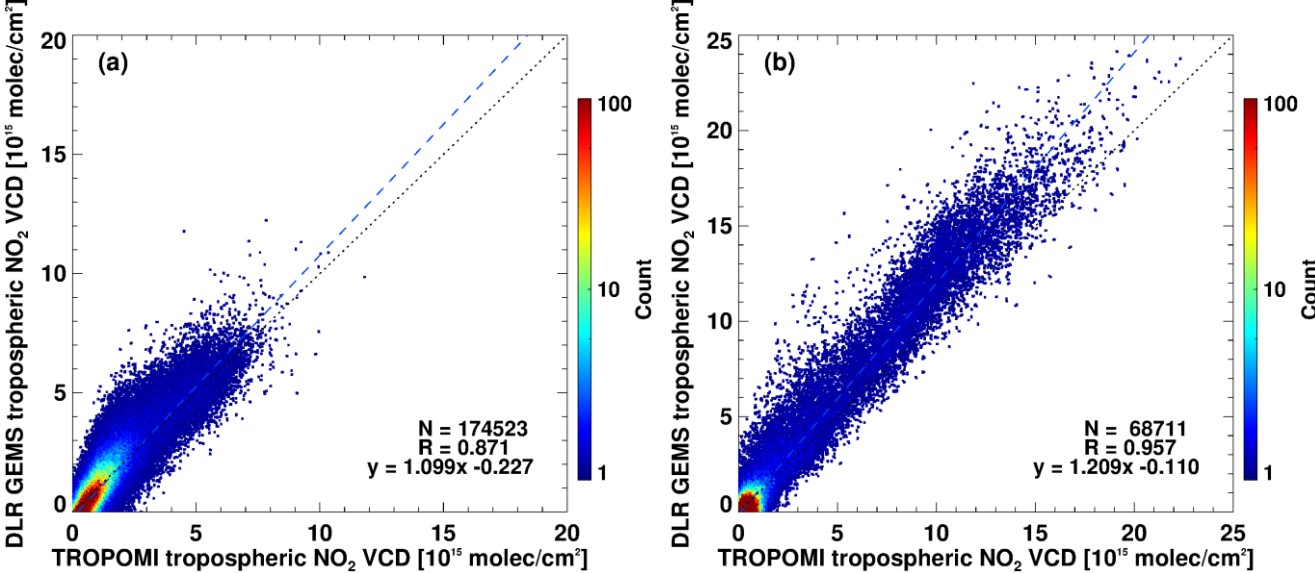

**Figure 20: Scatter plots of daily gridded DLR GEMS and TROPOMI v2.4 tropospheric NO₂ columns for (a) 14 June 2023 (displayed in Fig. 18) and (b) 14 December 2022 (displayed in Fig. 19), respectively. The color-coded points represent data density, and the blue dashed line indicates the linear regression.**


### 3.3 Comparison with GEMS v2.0 NO₂ product

The DLR GEMS and GEMS operational v2.0 NO₂ products can be directly compared on a pixel-by-pixel basis since they originate from identical GEMS L1 radiance and irradiance spectra. A comparative evaluation is performed using the regridded mean NO₂ columns for each scan time from selected four specific days in March, June, September, and December, for both

the DLR GEMS and GEMS operational v2.0 L2 products.

While the two retrieval algorithms are similar in that they largely follow three steps, DOAS-based NO₂ slant column retrievals, stratosphere-troposphere separation, and AMF calculation based on pre-calculated LUT, detailed differences in many aspects exist as described in previous sections. These differences include spectral fitting settings, stratospheric separation approaches, and ancillary data inputs in AMF calculation. Despite these differences, the final outputs of tropospheric NO₂ vertical columns

between the two products show overall good agreement with a high correlation ranging from 0.80 to 0.92 (Fig. 21). However, the DLR GEMS tropospheric NO₂ vertical column is $1.5 \times 10^{15}$ molec cm$^{-2}$ lower than the GEMS v2.0 tropospheric NO₂ columns on overage. The negative biases in tropospheric NO₂ columns between the DLR GEMS and GEMS v2.0 are also confirmed when referenced against the TROPOMI operational v2.4 product. When comparing GEMS v2.0 with TROPOMI v2.4 tropospheric NO₂ columns using the same methodology as described in Sect. 3.2, GEMS v2.0 exhibits values that are 20

– 60 % higher than those of TROPOMI, whereas DLR GEMS shows values only 10 – 20 % higher.





The differences in tropospheric $NO_2$ columns between the DLR GEMS and GEMS v2.0 retrieval algorithms may be influenced by several factors. One major factor contributing to the higher tropospheric $NO_2$ columns in GEMS v2.0 is the relatively low estimation of stratospheric $NO_2$ columns. In the GEMS operational v2.0 algorithm, stratosphere-troposphere separation is conducted based on the method proposed by Bucsela et al. (2013). Tropospheric $NO_2$ contribution is estimated using the WRF-

Chem model, and the a priori tropospheric $NO_2$ slant column density is subtracted from the total $NO_2$ slant column density to obtain an initial stratospheric vertical column. The field is masked wherever tropospheric contamination exceeds a pre-set threshold ($0.3\times10^{15}$ molec cm$^{-2}$), and then these estimated stratospheric vertical columns are binned onto a GEMS geographic grid. The binned vertical columns are interpolated over the masked areas, and the eliminated areas (hot spots) are smoothed and interpolated (Park et al., 2020). The stratospheric $NO_2$ vertical columns estimated by the GEMS operational v2.0 algorithm

show a range of $0.5\times10^{15}$–$1.5\times10^{15}$ molec cm$^{-2}$ for the 04:45 UTC scan time in June. This range is lower than the stratospheric $NO_2$ column ranges reported in previous studies (Dirksen et al., 2011; Belmonte Rivas et al., 2014) and the DLR GEMS stratospheric $NO_2$ columns, which range from $2.0\times10^{15}$ to $4.0\times10^{15}$ molec cm$^{-2}$ at the same scan time in June. Consequently, the lower values of stratospheric $NO_2$ columns in GEMS v2.0 lead to higher tropospheric $NO_2$ columns. It is anticipated that the forthcoming GEMS operational v3.0 $NO_2$ product will include updates to address this issue by correcting the low estimates

of stratospheric $NO_2$ column retrieval.

Another significant factor causing the difference in tropospheric $NO_2$ columns between DLR GEMS and GEMS v2.0 is the tropospheric AMF calculation. Firstly, the ancillary inputs used for AMF calculations in the two algorithms differ, and the effects of these differences in input datasets can manifest in various ways. In the DLR GEMS $NO_2$ retrieval algorithm, OCRA GEMS cloud fraction is applied instead of GEMS L2 cloud fraction. As discussed in Sect. 2.3.1.3, the OCRA cloud fraction

values in clear-sky scenes are lower, resulting in higher tropospheric AMFs and lower tropospheric $NO_2$ vertical columns compared to results obtained when applying cloud corrections using the GEMS L2 cloud fraction. In addition, $NO_2$ model profile shapes are different, as the DLR GEMS algorithm employs the CAMS forecast (CAMS IFS Cy48R1) a priori $NO_2$ profile available every 3 hours, while GEMS v2.0 utilizes the monthly mean hourly $NO_2$ profiles simulated from GEOS-Chem v13 (Park et al., 2020). Moreover, differences in the number and value of reference points in the altitude-dependent AMF LUT

between two algorithms can cause systematic biases.



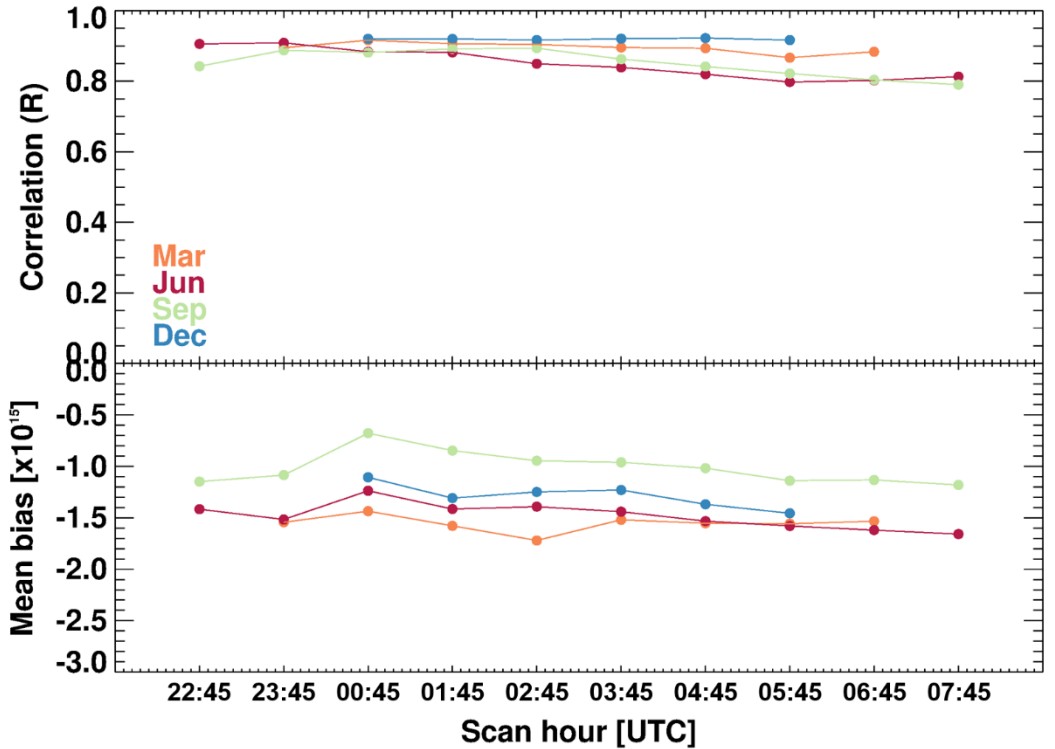

**Figure 21: Diurnal and seasonal variations in statistics (correlation and mean bias) between DLR GEMS and GEMS v2.0 tropospheric NO₂ columns. Tropospheric NO₂ VCDs from days 1, 10, 20, and 30 of March 2023, June 2023, September 2022, and December 2022 from both algorithms were used for the analysis.**


## 3.4 Uncertainty estimates for DLR GEMS tropospheric NO₂ vertical columns

The overall uncertainty of the tropospheric NO₂ column is determined through uncertainty propagation at each main retrieval step, which is performed independently and assumed to be uncorrelated (Boersma et al., 2004). The primary sources for uncertainty are (1) the signal-to-noise ratio of the radiance and irradiance measurements and spectral fitting errors affecting

the slant columns, (2) inaccuracies associated with the separation of stratospheric and tropospheric NO₂ columns, and (3) systematic errors arising from uncertainties in input parameters such as clouds, surface albedo, and a priori profile during AMF calculation.

The slant column uncertainty for DLR GEMS NO₂, estimated following a statistical method described in Sect. 2.1, is $0.9 \times 10^{15}$ molec cm$^{-2}$. The uncertainty in the stratospheric columns is estimated on average $2.0 \times 10^{14}$ molec cm$^{-2}$ based on the standard

deviation of model biases (differences between simulated CAMS NO₂ column and observed GEMS NO₂ column described in Sect. 2.2.2). The tropospheric AMF calculation is the largest source of uncertainties in tropospheric NO₂ vertical column retrievals, particularly for polluted conditions. The uncertainties on the tropospheric AMF are mainly dependent on



uncertainties of input parameters, such as the surface albedo $\alpha_s$, cloud top pressure $p_c$, cloud fraction $f_c$, and profile shape $p_h$, as well as on the sensitivity of the AMF to each of these parameters. The tropospheric AMF errors are calculated based on

uncertainty propagation through the squared sum of each parameter contribution as follows (Boersma et al., 2004; De Smedt et al., 2008):

$$\sigma_{AMF}^2 = \left(\frac{\partial AMF}{\partial \alpha_s}\sigma_{\alpha_s}\right)^2 + \left(\frac{\partial AMF}{\partial p_c}\sigma_{p_c}\right)^2 + \left(\frac{\partial AMF}{\partial f_c}\sigma_{f_c}\right)^2 + \left(\frac{\partial AMF}{\partial p_h}\sigma_{p_h}\right)^2 \tag{6}$$

The uncertainties of input parameters are typically estimated from the studies or obtained from comparisons with independent data. Due to missing information on the GEMS operational product for surface and cloud uncertainties, typical uncertainties used in previous studies are applied here ($\sigma_{\alpha_s}$= 0.02, $\sigma_{p_c}$= 50 hPa, $\sigma_{f_c}$= 0.05) (De Smedt et al., 2018; Song et al., 2021). The

uncertainty contribution from the a priori $NO_2$ profile $\sigma_{p_h}$ is effectively described by a profile height, defined as the altitude below which 75 % of the integrated $NO_2$ profile resides (De Smedt et al., 2018).

The AMF is very sensitive to the cloud top pressure when the cloud is located below or at the level of the $NO_2$ peak. The uncertainties arising from the cloud pressure can be up to 60 %, particularly for thick clouds located close to the boundary layer over polluted regions. For higher clouds with lower cloud top pressures, the sensitivity of the AMF is much weaker.

Cloud fraction also plays a significant role in determining the AMF and its associated uncertainty. While satellite measurements are filtered out for a cloud radiance fraction larger than 0.5 or a cloud fraction exceeding 0.3 in tropospheric $NO_2$ retrievals, the uncertainties related to cloud fraction are by ~25 %. The AMF sensitivity to albedo is relatively higher over polluted regions with profile shapes characterized by high surface $NO_2$ concentrations, compared to open ocean or clean background regions exhibiting smaller variations in profile shape. Large uncertainties of the AMF due to surface albedo are

mainly observed in cases where the input surface albedo values are different from actual surface conditions, especially for sudden snow or ice cover. In the case of the Asia GEMS domain, the impact of aerosols on the AMF sensitivity is an important factor to be considered regarding the particle properties and the vertical distribution. The aerosol effect on the AMF is not explicitly considered in the DLR GEMS $NO_2$ algorithm, as it is assumed that the effect of the non-absorbing part of the aerosol extinction is implicitly included in the cloud correction through the effective cloud parameters (Boersma et al., 2004, 2011).

The overall uncertainty in DLR GEMS tropospheric $NO_2$ vertical columns varies depending on observation scenarios. In clean background regions such as open ocean and rural areas, the retrieval uncertainties typically range on overage 10 - 30 %, primarily dominated by slant column uncertainties. On the other hand, in polluted regions with high tropospheric $NO_2$ abundances, uncertainties in tropospheric $NO_2$ vertical columns are largely affected by AMF errors. Particularly in heavily polluted areas during winter, where low-level clouds are present (with high cloud top pressure), the total uncertainty in

tropospheric $NO_2$ columns is most pronounced, reaching up to 60 %.



## 4. Summary and conclusions

The geostationary satellite instrument GEMS performs high-resolution observations in both time and space, enabling the monitoring of diurnal variability in atmospheric compositions over Asia. In this study, we developed an advanced retrieval algorithm for tropospheric $NO_2$ columns from geostationary satellite spectrometers, and applied it to GEMS measurements. The DLR GEMS $NO_2$ retrieval algorithm follows the heritage from previous and existing algorithms used for the GOME-2 and TROPOMI instruments, but improved approaches are applied to reflect the specific features of geostationary satellites, such as high temporal samplings, limited range of spatial coverage, larger zenith angles and high spatial resolution.

The DLR GEMS $NO_2$ retrieval process begins with slant column retrievals based on the DOAS method. This algorithm employs a fitting window of 425-480 nm, which is extended compared to the current fitting window used in GEMS operational v2.0 algorithm, to avoid saturation effects (primarily occurring over bight clouds in equatorial regions) at shorter wavelengths and systematic spectral features observed above 480 nm. In addition to trace gas absorption cross-sections, including $NO_2$ and related interfering absorbers, a pseudo cross-section is introduced in the DOAS fit for polarization correction. DLR GEMS $NO_2$ slant columns show improved quality and lower uncertainties compared to those from GEMS L2 v2.0, as assessed through a posteriori statistical analysis. These improvements in GEMS $NO_2$ slant columns are mainly attributed to the use of a larger fitting window, along with minor effects with polarization correction and advanced spectral calibration.

The total slant column retrieved by the DOAS fit is separated into stratospheric and tropospheric contributions. For the stratosphere-troposphere separation in GEMS measurements, we developed and evaluated two approaches: (1) STREAM, originally employed for stratosphere-troposphere separation in current polar orbiting satellites, adapted to geostationary satellite observations, (2) estimation of stratospheric $NO_2$ columns using the CAMS forecast (IFS Cy48R1) model, which introduced comprehensive stratospheric chemistry. Both methods successfully provide hourly estimates of stratospheric $NO_2$, exhibiting diurnal variability of stratospheric fields within reasonable value ranges. However, STREAM has limitations in describing small-scale variations in stratospheric $NO_2$ due to the application of a relatively coarse convolution kernel, and systematic biases occurred near the boundary of FOV due to limited domain coverage. On the other hand, the stratospheric $NO_2$ column estimated from the CAMS forecast model not only describes the diurnal cycle but also captures the small-scale variations in stratospheric $NO_2$ fields, which demonstrates that this approach is more suitable for application to geostationary missions like GEMS, TEMPO and Sentinel-4 with high spatiotemporal resolution. It is notable that when utilizing CAMS forecast model data for stratospheric $NO_2$ estimation, biases between the modelled columns and observed GEMS columns should be corrected. In this algorithm, model bias patterns in the stratosphere are parameterized as a function of latitude and scan hour using selected stratospheric dominant pixels, such as clouded or clean background pixels.

Tropospheric $NO_2$ slant columns are converted to vertical columns by applying the tropospheric AMF. In the tropospheric AMF calculation, cloud properties, surface albedo and a priori $NO_2$ profiles are important, as they significantly affect the tropospheric AMF values. In this study, we evaluated the influence of cloud correction on GEMS tropospheric $NO_2$ retrieval using two different cloud fraction datasets. One is cloud fractions retrieved from DLR OCRA retrieval adapted to GEMS level



1 data, while the other is cloud fractions from GEMS L2 v2.0 product. Although the two products have overall good agreement,
spatial patterns are found that OCRA cloud fractions are smaller by about 0.1 than GEMS L2 v2.0 cloud fractions for clear-sky scenes. Compared to cloud corrections using the GEMS L2 v2.0 cloud fraction, the use of smaller OCRA cloud fractions increases the tropospheric AMF by ~28 % and decreases the tropospheric $NO_2$ columns by ~22 % for clear-sky conditions in polluted regions. Additionally, when cross-comparison with the TROPOMI product for clear-sky conditions, OCRA cloud fractions and tropospheric $NO_2$ columns retrieved using OCRA-based cloud corrections show closer agreement compared to
GEMS L2 v2.0 cloud fractions and the applied retrieval results.

The impacts of surface albedo on GEMS tropospheric $NO_2$ retrievals were assessed by comparing the GEMS v2.0 BSR and TROPOMI LER climatology v2.0 product. In general, the GEMS BSR shows higher values over land and lower values over the sea in comparison to the TROPOMI LER climatology. The difference between the two products is more apparent in winter, which may be attributed to discrepancies between the TROPOMI LER climatology and actual surface conditions, particularly
in snow/ice scenes. The differences in tropospheric $NO_2$ columns depending on surface albedo are more pronounced in winter than in summer, especially over heavily polluted regions such as eastern China and northern India compared to clean background regions.

The a priori $NO_2$ profiles from the CAMS forecast model, applied in the DLR GEMS algorithm, effectively capture variations in $NO_2$ concentrations depending on emission patterns and meteorological conditions throughout the day with a high spatial
and temporal resolution. CAMS forecast a priori $NO_2$ profiles show the highest surface concentration during morning commuting hours with elevated traffic volumes. As noon approaches, $NO_2$ concentrations decrease due to reduced source emissions, enhanced vertical mixing, and photolysis. The minimum tropospheric $NO_2$ concentration during midday begins to increase again with increased traffic towards the end of work hours. This highlights the suitability of CAMS forecast a priori profiles for tropospheric $NO_2$ retrievals from geostationary satellite measurements.

The retrieved DLR GEMS tropospheric $NO_2$ columns show the spatial and temporal variability over Asia with high spatial resolution. High tropospheric $NO_2$ columns are observed over urban and industrial regions in eastern China, South Korea, and northern India, with hotspot signals detected in major city clusters. Tropospheric $NO_2$ levels peak in the morning in urban areas due to commuter traffic emissions, decrease around noon, and then rise again in the afternoon. In addition, spatial gradients from city centers to surrounding areas are detected, with high $NO_2$ levels in megacities gradually decreasing with
distance due to reduced emissions and horizontal transport. In open ocean and rural regions, where $NO_2$ sources are primarily natural emissions, significantly lower tropospheric $NO_2$ columns are observed.

The evaluation of DLR GEMS tropospheric $NO_2$ columns was performed by comparisons with independent TROPOMI L2 v2.4 and GEMS L2 v2.0 $NO_2$ products. Spatiotemporally regridded DLR GEMS and TROPOMI $NO_2$ data were used for comparisons. Overall, good agreement is observed between DLR GEMS and TROPOMI tropospheric $NO_2$ columns with high
correlation coefficients (R = 0.87 and 0.96 for test days). The datasets also agree well in terms of spatial distribution and value range of tropospheric $NO_2$. However, systematic biases are evident, with GEMS tropospheric $NO_2$ columns showing lower values in the open ocean and higher values over land, particularly in polluted regions or at the edges of the domain. These





differences are attributed to systematic gradients in GEMS $NO_2$ slant columns in the north-south direction (arising from the imperfect radiometric calibration and residual corrections in GEMS L1 v1.2.4) and tropospheric AMFs calculated using

different ancillary input datasets. Additionally, the comparison between DLR GEMS and GEMS L2 v2.0 $NO_2$ also shows good agreement with high correlation coefficients ranging from 0.80 to 0.92. However, DLR GEMS tropospheric $NO_2$ columns consistently show lower values compared to GEMS v2.0, with mean biases of $1.5 \times 10^{15}$ molec $cm^{-2}$. The differences may be due to significantly underestimated stratospheric $NO_2$ columns in GEMS v2.0 and differences in ancillary inputs used in AMF calculations.

The overall uncertainty in DLR GEMS tropospheric $NO_2$ vertical columns varies depending on the observation conditions. In regions with minimal pollution, like open ocean and rural areas, retrieval uncertainties typically range from 10 % to 30 %, mainly driven by uncertainties in slant columns. Conversely, in heavily polluted regions with high tropospheric $NO_2$ levels, uncertainties in tropospheric $NO_2$ columns are primarily influenced by errors in tropospheric AMF calculations. In winter, especially in heavily polluted areas with low-level clouds, the total uncertainty in tropospheric $NO_2$ columns is most

pronounced.

The DLR GEMS $NO_2$ data quality will be further analysed using additional data from ground-based measurements including Multi-Axis differential optical absorption spectroscopy (MAX-DOAS), Pandora, and in-situ measurements covering different pollution conditions and periods. In addition, based on the good capability of the DLR GEMS tropospheric $NO_2$ columns to capture the temporal and spatial variability at the city scale, estimations of the diurnal cycle in $NO_2$ source emissions over Asia

will be further studied. Remaining issues contributing to retrieval uncertainties and systematic biases in the current version of the DLR GEMS $NO_2$ algorithm will be improved in the near future with the application of updated GEMS operational L1 and L2 version data, set for release in the near future. Furthermore, this retrieval algorithm can be easily adapted to other geostationary satellite instruments, such as TEMPO and Sentinel-4, enabling hourly monitoring and analysis across each continent in the future.





*Data availability.* The DLR GEMS NO₂ product is available on request. The DLR GEMS OCRA product is available on request. GEMS L2 v2.0 NO₂ and cloud product can be accessed at https://nesc.nier.go.kr/en/html/cntnts/91/static/page.do (National Institute of Environmental Research, NIER, last access: 15 April 2024). TROPOMI NO₂ product are freely available via https://s5phub.copernicus.eu/ (Sentinel-5P Pre-Operations Data Hub, last access: 15 April 2024). TROPOMI directionally dependent Lambertian-equivalent reflectivity (DLER) product is freely available via https://www.temis.nl/surface/albedo/tropomi_ler.php (TEMIS website, last access: 15 April 2024).


*Author contributions.* SS designed and developed the DLR GEMS NO₂ algorithm. SS, PV, and DL contributed to the study conception and design. RL provided the DLR GEMS OCRA product. KPH, PH, and PV collected datasets and assisted in data interpretation. HL and JK provided information on the GEMS L2 products and algorithms. SS performed the final data analysis and interpreted the results together

with co-authors. SS wrote the paper with feedback and contributions from all co-authors.

*Competing interests.* At least one of the (co-)authors is a member of the editorial board of Atmospheric Measurement Techniques.


*Acknowledgements.* We thank the National Institute of Environmental Research of South Korea for providing GEMS data. The authors also acknowledge the support and information from the PEGASOS project (funding by ESA under ESA Contract No. 4000138176/22/I-DT-lr). The authors also gratefully acknowledge the free use of Sentinel-5 Precursor data. Sentinel-5 Precursor is a European Space Agency (ESA) mission on behalf of the European Commission (EC).



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
