# Peer review of "Tropospheric NO2 retrieval algorithm for geostationary satellite instruments: applications to GEMS"

_EGUsphere, 2024_

## Author Comment (AC1)

**Response to Referee #1**

We gratefully appreciate the reviewer for the careful reading of our manuscript and for the very constructive comments. We were able to enhance the scientific quality of our manuscript by incorporating the reviewer's comments and suggestions. Below, the reviewer's text is given in black while our replies and descriptions on how the comments have been addressed in the manuscript are given in blue.

The manuscript by Seo et al. describes the application of the DOAS algorithm to the GEMS visible spectra to retrieve tropospheric $NO_2$ vertical columns. Compared to the current (v2.0) operational $NO_2$ retrieval from GEMS, The DOAS implementation by Seo et al. included significant improvements in slant $NO_2$ column estimations and their conversions into vertical columns. This manuscript is well-written, and I recommend its publication. I have a few comments that the authors may address to help a reader better appreciate the algorithm implementation.

1. The authors used a broader spectral range (425 - 480 nm) to improve the slant column estimation, particularly in reducing its noise level compared to that from the narrower range (432 - 450 nm) used by the operational algorithm. However, the manuscript contains little discussion about the slant column biases. How do the biases from the broader window compare with those of the narrower window of the operational GEMS $NO_2$? Are the biases higher, lower, or similar in magnitudes and north-south behavior?

The GEMS operational v2.0 $NO_2$ slant columns show a more pronounced north-south gradient with higher slant column uncertainties compared to the DLR $NO_2$ slant columns (Fig. R1.1). However, it is difficult to conclude that the higher slant column biases along the north-south direction are solely due to the smaller spectral fitting window, as there are differences in other spectral fitting settings, including polarization correction, intensity offset correction, and absorption cross sections. To precisely investigate the effect of the fitting window on GEMS $NO_2$ slant column retrievals, we compared the spatial distributions of slant columns by changing the fitting window while keeping other reference spectral fit settings (including polarization correction and intensity offset correction) constant. As shown in Fig. R1.2, the GEMS $NO_2$ slant columns derived using the narrower fitting window of 432-450nm is higher than those obtained using the broader fitting window of 425-480 nm. However, the slant column biases in the north-south direction do not differ significantly between the two fitting windows.

[Figure]

Figure R1.1. Maps of GEMS $NO_2$ slant columns from (a) the DLR spectral fitting retrieval (presented in this study) and (b) GEMS operational v2.0, and (c) the differences between two (b - a) for 15 March 2023 at 04:45 UTC.

[Figure]

Figure R1.2. Maps of GEMS NO$_2$ slant columns retrieved using the DLR reference spectral fitting settings with a fitting window of (a) 425 – 480 nm (used in this study) and (b) 432 – 450 nm, and (c) the differences between the two (b - a) for 15 March 2023 at 04:45 UTC.

2. The broader spectral range (425 - 480 nm) includes possible soil signatures (Richter et al., 2011). Does slant column fitting include a soil signature term over areas where it may be present?

We did not detect any specific residual features related to soil over desert or bare soil regions within the GEMS field of view from the current spectral range of 425 – 480 nm. Nevertheless, we investigated the effect of soil signals on the GEMS NO$_2$ spectral fitting by applying an empirical soil signature as detailed in Richter et al. (2011) (refer to Fig. 6 therein). Figure R2.1 shows the spatial distribution of the soil signal retrieved using this empirical soil signature in the GEMS NO$_2$ spectral fitting. The fitted soil signals exhibit rather unphysical and random distributions, with positive values over the southern FOV including the ocean and lands below ∼ 35 °N, and negative values over the northern part of the FOV. This soil signature is reflected in the NO$_2$ slant column retrievals as shown in Fig. R2.2, which does not lead to a physically meaningful improvement in NO$_2$ slant column retrievals in current DLR GEMS spectral fitting settings.

[Figure]

Figure R2.1. The spatial distribution of the soil signal retrieved by applying the empirical soil signature derived by Richter et al. (2011) in the DLR GEMS NO$_2$ spectral fitting.

[Figure]

Figure R2.2. GEMS geometric NO$_2$ vertical columns retrieved using (a) the standard setting (without soil signature) and (b) the setting with the additional inclusion of the soil, and (c) the differences between the two (b-a) for 15 March 2023 at 04:45 UTC.

3. Eq. (1) (page 4) contains the offset term (offset($\lambda$), a linear function of wavelength $\lambda$). Please describe the impact on the slant column estimation. Does its inclusion reduce the noise level of the slant column or change (consistently increase or decrease) the slant columns to reduce biases? In short, please justify this offset term.

As suggested by the reviewer, we have added a more detailed analysis of the impact of the intensity offset correction on GEMS NO$_2$ slant column retrievals in Appendix A, which will be included in the revised version as below.

Section 2.1: "An intensity offset correction is fitted as an additional linear term to correct issues related to incomplete removal of straylight in the spectrometer, inelastic scattering in the ocean water, and dark current in the level 1 spectrum (Platt and Stutz, 2008; Richter et al., 2011). *The effect of the intensity offset correction on GEMS NO$_2$ slant column retrievals is described in more detail in Appendix A1.*"

Appendix A: Spectral fit sensitivity studies

We investigate the effect of different spectral fit settings on the GEMS NO$_2$ slant column retrievals. Since the slant column is influenced by variations in viewing geometries, such as solar zenith angle (SZA) and viewing zenith angle (VZA), we used the geometric NO$_2$ vertical column, obtained by dividing the slant column by the geometric AMF (1/cos(SZA) + 1/sin(VZA)), to facilitate evaluations and comparisons.

A1. Intensity offset correction

As described in Sect 2.1, the intensity offset correction is applied in the standard setting for DLR GEMS NO$_2$ slant column retrievals in this study. We evaluated the impact of this correction by comparing GEMS NO$_2$ spectral fitting retrievals with and without the intensity offset correction, while keeping other settings identical.

Although the overall impact of the intensity offset correction is small, its application results in increased slant columns over the ocean located south of the FOV (Fig. A1). This increase is attributed

to the partial compensation of vibrational Raman scattering in ocean water through the additive offset correction in the spectral fitting. The increased slant columns over the ocean due to the intensity offset correction mitigate the known systematic biases along the north-south direction of GEMS $NO_2$ slant columns (see Sect. 3.2). Additionally, the inclusion of the intensity offset correction in GEMS $NO_2$ spectral fitting retrievals leads to a decrease in fitting RMS values and slant column uncertainties, though the improvement is minor.

[Figure]

Figure A1. GEMS geometric $NO_2$ vertical columns retrieved (a) with and (b) without intensity offset correction, and (c) the relative differences between the two ((b – a)/a) for 15 March 2023 at 04:45 UTC.

4. On Page 5, line 141, a pseudo-cross-section for polarization correction is added to the slant column fitting. Please describe its impact on the slant column.

As suggested by the reviewer, we have added a more detailed analysis of the impact of the polarization correction on GEMS $NO_2$ slant column retrievals in Appendix A, which will be included in the revised version.

A2. Polarization correction

To improve the retrieval of trace gases from satellite measurements, it is important to obtain stable and precise measurements of reflected radiance. One of the error sources in the measured radiance spectrum is the polarization of light. The radiometric response of a satellite instrument is influenced by the polarization state of incoming light, which is caused by gratings, mirrors, and prisms. To reduce the instrument's polarization sensitivity, two representative methods have been used: (1) a depolarization method that destroys the polarization information through scrambling, and (2) a detection of polarization states in the atmosphere using specialized devices. However, since GEMS lacks both a polarization measurement device (PMD) and a scrambler, polarization correction should be applied to improve the spectral fitting retrieval of $NO_2$ from GEMS measurements (Choi et al., 2024). Therefore, we included a pseudo cross-section to correct the spectral polarization sensitivity of GEMS as described in Sect. 2.1.

Figure A2 illustrates the impact of polarization correction on the GEMS $NO_2$ column retrieval by comparing results with and without polarization correction under identical spectral fitting settings. Without polarization correction, inhomogeneous spatial distributions of $NO_2$ slant columns are observed. Notably, the uneven $NO_2$ slant column distribution is closely related to cloud distribution over the ocean, as scattering in liquid/ice clouds and reflection at the dark ocean surface significantly influence the degree of polarization. In addition, we found a diurnal variation in the effect of

polarization correction on NO$_2$ slant column retrievals. Polarization sensitivity is strongly influenced by changes in viewing geometry, particularly the change of the solar zenith angle (larger solar zenith angles result in greater polarization sensitivity, while smaller solar zenith angles result in lower polarization sensitivity). Therefore, polarization correction has a larger effect on NO$_2$ slant column retrievals in the early morning and sunset compared to noon (see Fig. A3).

[Figure]

Figure A2. GEMS geometric NO$_2$ vertical columns retrieved (a) with and (b) without polarization correction, and (c) the differences between the two (b - a) for 15 March 2023 at 04:45 UTC.

[Figure]

Figure A3. The diurnal variation of the difference in GEMS geometric NO$_2$ vertical columns without and with polarization correction on 5 March 2023 from 00:45 to 06:45 UTC.

5. On Page 10, line 267, a low-order bivariate polynomial is mentioned. Figure 5 suggests that it is only a single variable (i.e., latitude) polynomial. Looks like the UTC is a label only. Since the polynomial is time-dependent, the polynomial should indeed be bivariate, with the second variable being the longitude. However, the time difference is probably sufficiently small that the longitudinal dependence may be neglected.

Thank you for pointing out the misleading expression. As you suggested, we revised the texts as follows:

"Assuming that model biases for the stratospheric fields depend on latitude and observation time (local time of day), the weighted biases for respective pixels are fitted with a low-order polynomial as a function of latitude for each GEMS scan hour (see Fig. 5)."

As you mentioned, although the model bias correction term is time-dependent, the range of the model bias correction values did not vary significantly over time. Therefore, it was more efficient to derive a latitude-dependent model bias correction term for each scan hour, rather than computing a daily polynomial fit for model bias correction using both latitude and scan hour as variables.

6. The authors used the OCRA cloud fraction in AMF calculations, avoiding the cloud fractions from the GEMS standard product, which likely biased high due to high (likely ~8% or higher) biases in GEMS sun-normalized radiances. However, the authors selected GEMS (v2.0) cloud pressure and GEMS (v2.0) background surface reflectance (BSR) for AMF calculations. Please comment on the possible biases in these products due to radiance biases.

The DLR OCRA algorithm for cloud fraction retrieval uses instrument L1b radiance, irradiance and viewing geometry as input data (Loyola et al., 2018). In this study, the cloud fraction retrieval is obtained by OCRA algorithm adapted to the GEMS L1 radiance and irradiance data. The DLR cloud retrieval for cloud pressure (and surface properties as auxiliary input) with the DLR ROCINN algorithm requires the Oxygen A-band in the NIR. Since this wavelength range is not covered by the GEMS instrument, we used the cloud pressure and surface reflectance from the GEMS CLD and BSR products in this study.

The high bias in GEMS v2.0 cloud fraction compared to OCRA cloud fraction for clear-sky scenes might be attributed to different treatments of surface albedo in cloud retrievals. The GEMS v2.0 cloud retrieval algorithm uses surface reflectance climatology from OMI as input (Kim et al., 2024). We found the surface features from OMI-based surface LER climatology (particularly for bright surfaces) get translated into the operational GEMS L2 cloud fraction product (see Fig. R3 below), which is the main reason why we used the OCRA cloud fraction instead of the GEMS cloud fraction. A detailed verification of the GEMS cloud pressure and GEMS surface reflectance and the impact of possible reflectance biases on those is however out of the scope of this manuscript.

[Figure]

Figure R3. Zoom into a clear-sky region over China on the 2:45 UTC scan on 5 June 2021 for the OCRA cloud fraction (top left) and the operational GEMS v2.0 L2 cloud fraction (top right). The bottom left panel shows a true-colour RGB from VIIRS, taken from NASA Worldview. Note that the bright surface structures appearing in the northern part of the RGB true colour image seem to be translated into the operational GEMS cloud fraction product.

**References**

Choi, H., Liu, X., Jeong, U., Chong, H., Kim, J., Ahn, M. H., Ko, D. H., Lee, D.-W., Moon, K.-J., and Lee, K.-M.: Geostationary Environment Monitoring Spectrometer (GEMS) polarization characteristics and correction algorithm, Atmos. Meas. Tech., 17, 145–164, https://doi.org/10.5194/amt-17-145-2024, 2024.

Kim, B.-R., Kim, G., Cho, M., Choi, Y.-S., and Kim, J.: First results of cloud retrieval from the Geostationary Environmental Monitoring Spectrometer, Atmos. Meas. Tech., 17, 453–470, https://doi.org/10.5194/amt-17-453-2024, 2024.

Loyola, D. G., Gimeno García, S., Lutz, R., Argyrouli, A., Romahn, F., Spurr, R. J. D., Pedergnana, M., Doicu, A., Molina García, V., and Schüssler, O.: The operational cloud retrieval algorithms from TROPOMI on board Sentinel-5 Precursor, Atmos. Meas. Tech., 11, 409–427, https://doi.org/10.5194/amt-11-409-2018, 2018.

Richter, A., Begoin, M., Hilboll, A., and Burrows, J. P.: An improved NO 2 retrieval for the GOME-2 satellite instrument, Atmos. Meas. Tech., 4, 1147–1159, https://doi.org/10.5194/amt-4-1147-2011, 2011.

---

## Author Comment (AC3)

**Response to Referee #2**

We gratefully appreciate the reviewer for the careful reading of our manuscript and for the very constructive comments. We were able to enhance the scientific quality of our manuscript by incorporating the reviewer's comments and suggestions. Below, the reviewer's text is given in black while our replies and descriptions on how the comments have been addressed in the manuscript are given in blue.

The authors describe a new algorithm to retrieve $NO_2$ from the GEMS satellite instrument. The algorithm is based on what has been used for polar-orbiting satellites and has been modified for application to geostationary satellites. The authors evaluate the retrieved $NO_2$ columns using TROPOMI and the operational GEMS retrievals and discuss the uncertainties in the retrievals.

This is a high-quality manuscript. It includes a complete description of the new algorithm and a thorough analysis of its strengths and weaknesses. It is well-written and makes good use of figures. I point out below a few points that could be better addressed.

1. The reader would benefit from a brief description of the operational GEMS algorithms near the beginning of the paper. It would help to include a table describing the major similarities and differences between the new and the operational GEMS algorithm.

Thanks for your suggestion. As the reviewer recommended, we will include Table 1 in Sect. 2 of the revised manuscript to summarize the main settings of the DLR GEMS and the operational v2.0 GEMS $NO_2$ algorithm.

Table 1. Overview of the GEMS tropospheric $NO_2$ retrievals from the DLR (this study) and GEMS operational v2 algorithm.

| | | DLR GEMS (this work) | GEMS operational v2.0 (Park et al., 2020) |
|---|---|---|---|
| Spectral fit settings for slant column retrievals | Fitting window | 425 – 480 nm | 432 - 450 nm |
| | Reference spectrum | Daily solar irradiance | Daily solar irradiance |
| | Absorption cross - sections | $NO_2$, $O_3$, $H_2O_{vap}$, $H_2O_{liq}$, $O_4$ | $NO_2$, $O_3$, $H_2O_{vap}$, $O_4$ |
| | Pseudo absorbers | Ring, Polarization sensitivity | Ring |
| | Polynomial | Fourth order | Second order |
| Stratosphere-troposphere separation | | Stratospheric $NO_2$ estimation based on the CAMS forecast IFS Cy48R1 profile (detailed in Sect. 2.2.2) | Approach based on Bucsela et al. (2013) |
| Auxiliary input parameters for AMF calculations | Cloud parameter | Cloud fraction: OCRA adapted to GEMS

Cloud pressure: GEMS cloud v2.0 (Kim et al., 2024) | Cloud fraction and pressure from GEMS cloud v2.0 |

| | Surface albedo | GEMS BSR v2.0 (Sim et al., 2024) | GEMS BSR v2.0 |
|---|---|---|---|
| | A priori $NO_2$ profile | CAMS forecast IFS Cy48R1 (Eskes et al., 2024) | Monthly mean hourly $NO_2$ profiles simulated from GEOS-Chem v13 |

2. The uncertainty analysis (Section 3.4) could be improved by discussing uncertainties specific to retrievals from geostationary satellites. For example, how do the retrieval uncertainties vary with the time of day, or how different are they near the edge of the field of view compared to the center?

In this study, the total uncertainty in tropospheric $NO_2$ vertical columns is derived through the uncertainty propagation, which is composed of slant column uncertainties, stratospheric column uncertainties, and tropospheric AMF uncertainties (described in Sect. 3.4 and Eq. 6 in the revised manuscript).

The most critical and challenging aspect of temporal variations in the expected total error in tropospheric $NO_2$ columns is the calculation of tropospheric AMF uncertainties. Important error sources are related to the a priori tropospheric profile shape, surface directional reflectance, and the cloud correction. Since the uncertainties in the tropospheric AMF depend on the uncertainty of the input parameter ($\sigma_{parameter}$) and the sensitivity of the AMF to each parameter ($\frac{\partial M}{\partial parameter}$), the accuracy of the error analysis is significantly affected by the precision of the auxiliary input data uncertainties. However, as mentioned in the manuscript, due to the lack of information on the uncertainties of GEMS auxiliary data products (i.e., GEMS surface albedo (GEMS BSR), GEMS cloud fraction and cloud pressure), typical fixed parameter uncertainties ($\sigma_{\alpha_s}$= 0.02, $\sigma_{p_c}$= 50 hPa, $\sigma_{f_c}$= 0.05) used in previous TROPOMI studies were applied in this study. Therefore, we emphasize that a more precise analysis of total GEMS tropospheric $NO_2$ column uncertainties should be performed in the future once information on the GEMS auxiliary data uncertainties is available.

Despite the current limitations, we will add the following text about total uncertainty in GEMS tropospheric $NO_2$ columns for diurnal variation in Sect 3.4 (of the revised manuscript) as follows:

"In addition, the total uncertainty in GEMS tropospheric $NO_2$ columns varies with the scan hour (local time of the day). Generally, the total uncertainty is higher in polluted areas during the early morning rush hour. This is associated with a high peak-shaped profile near the surface due to increased traffic emissions within the lower atmospheric boundary layer. These shapes of a priori profiles for typical commuting hours lead to lower tropospheric AMFs ($M_{tr}$), resulting in higher total uncertainties in tropospheric $NO_2$ columns (see Eq. 6) compared to local noon.

Furthermore, for geostationary satellites observing from a fixed position, extreme viewing angles (near the edge of the scan) generally increase the slant column uncertainty due to higher spectral noise. However, this increased slant column uncertainty at large viewing geometries (high solar zenith angle and viewing zenith angle) does not always lead to higher total uncertainty in tropospheric $NO_2$, since the total uncertainty in tropospheric $NO_2$ column is also influenced by the tropospheric AMF, which varies not only with viewing geometry but also with other factors such as surface albedo and surface pressure."

3. The effect of aerosols on the $NO_2$ retrievals is important for Asia, and in most retrievals, it is considered implicitly in the cloud parameters retrieved using O2-O2 absorption. It is unclear whether the OCRA algorithm used in this work does the same. Does it instead correct for the presence of aerosols and could that partly explain why it retrieves lower cloud fractions (figures 9 and 10) compared to the GEMS retrieval?

As described in Sect. 2.3 and Table 1 (to be included in the revised manuscript), we used the cloud fraction from the OCRA algorithm (i.e., the operational TROPOMI (Loyola et al., 2018) and upcoming S4 cloud fraction retrieval algorithm) adapted to GEMS L1 data, and the cloud centroid pressure from the GEMS v2.0 L2 cloud product, which is based on the LUT approach using $O_2$-$O_2$ absorptions. The DLR cloud retrieval for cloud pressure with the ROCINN algorithm requires the Oxygen A-band in the NIR. Since this wavelength range is not covered by the GEMS instrument, we used the cloud pressure data from the GEMS v2.0 cloud product.

OCRA does not explicitly correct for the presence of aerosols. As for the parameters retrieved using the $O_2$-$O_2$ absorption, aerosols are also implicitly included in OCRA because OCRA retrieves a cloud fraction based on comparing the measured reflectance (irrespective if clouds or aerosols contribute to the reflectance) to the expected reflectance in fully clear and fully cloudy conditions. Hence, a very strong aerosol contamination could also be misinterpreted as a "false" cloud fraction by OCRA.

The differences seen for some clear scenes between OCRA and the GEMS cloud fraction product (see Fig. R1 below) are, to our understanding, more likely related to the different surface treatments in the two algorithms. According to Kim et al. (2024), the GEMS v2.0 cloud retrieval algorithm uses a monthly surface reflectance climatology from OMI as input, whereas OCRA uses a clear-sky climatology based on the EPIC instrument onboard the NASA DSCOVR platform. We found the surface features from OMI-based surface LER climatology (particularly for bright surfaces) get translated into the operational GEMS L2 cloud fraction product as shown in Fig. R1. This does not seem to be the case for OCRA when using the EPIC clear-sky climatology.

[Figure]

Figure R1. Zoom into a clear-sky region over China on the 2:45 UTC scan on 5 June 2021 for the OCRA cloud fraction (top left) and the operational GEMS v2.0 L2 cloud fraction (top right). The bottom left panel shows a true-colour RGB from VIIRS, taken from NASA Worldview. Note that the bright surface structures appearing in the northern part of the RGB true colour image seem to be translated into the operational GEMS cloud fraction product.

4. Equation 6: Is there an error correlation between albedo and cloud fraction (Boersma et al. 2018; doi: 10.5194/amt-11-6651-2018)?

In this study, as specified in Eq. (7) in Sect. 3.2 (revised version), we did not account for the contribution from the possible correlation between the cloud fraction and the surface albedo. We will consider the error correlation between cloud fractions and surface albedo in the tropospheric AMF uncertainty calculation in a follow-up study, using an improved version of GEMS surface albedo and cloud products and their uncertainties, which will be released in the near future.

5. Line 547-9: $NO_2$ is also photolyzed by visible radiation, not just UV. Another factor for the low noontime values of $NO_2$ is oxidation by OH.

Thank you for pointing out this. We revised the sentences (Line 547-549) as follows:

"During midday, given sufficient solar radiation, $NO_2$ is photolyzed to produce NO and oxygen atoms, and it is also oxidized by OH radicals, resulting in a decrease in tropospheric $NO_2$ levels."

**References**

Bucsela, E. J., Krotkov, N. A., Celarier, E. A., Lamsal, L. N., Swartz, W. H., Bhartia, P. K., Boersma, K. F., Veefkind, J. P., Gleason, J. F., and Pickering, K. E.: A new stratospheric and tropospheric NO 2 retrieval algorithm for nadir-viewing satellite instruments: applications to OMI, Atmos. Meas. Tech., 6, 2607–2626, https://doi.org/10.5194/amt-6-2607-2013, 2013.

Eskes, H., Tsikerdekis, A., Ades, M., Alexe, M., Benedictow, A. C., Bennouna, Y., Blake, L., Bouarar, I., Chabrillat, S., Engelen, R., Errera, Q., Flemming, J., Garrigues, S., Griesfeller, J., Huijnen, V., Ilic, L., Inness, A., Kapsomenakis, J., Kipling, Z., Langerock, B., Mortier, A., Parrington, M., Pison, I., Pitkanen, M., Remy, S., Richter, A., Schoenhardt, A., Schulz, M., Thouret, V., Warneke, T., Zerefos, C., and Peuch, V.-H.: Technical Note: Evaluation of the Copernicus Atmosphere Monitoring Service Cy48R1 upgrade of June 2023, EGUsphere [preprint], https://doi.org/10.5194/egusphere-2023-3129, 2024.

Kim, B.-R., Kim, G., Cho, M., Choi, Y.-S., and Kim, J.: First results of cloud retrieval from the Geostationary Environmental Monitoring Spectrometer, Atmos. Meas. Tech., 17, 453–470, https://doi.org/10.5194/amt-17-453-2024, 2024.

Loyola, D. G., Gimeno García, S., Lutz, R., Argyrouli, A., Romahn, F., Spurr, R. J. D., Pedergnana, M., Doicu, A., Molina García, V., and Schüssler, O.: The operational cloud retrieval algorithms from TROPOMI on board Sentinel-5 Precursor, Atmos. Meas. Tech., 11, 409–427, https://doi.org/10.5194/amt-11-409-2018, 2018.

Park, J., Lee, H., and Hong, H.: Geostationary Environment Monitoring Spectrometer (GEMS) Algorithm Theoretical Basis Document NO$_2$ Retrieval Algorithm, available at: https://nesc.nier.go.kr/ko/html/satellite/doc/doc.do (last access: 15 April 2024), 2020.

Sim, S., Choi, S., Jung, D., Woo, J., Kim, N., Park, S., Kim, H., Jeong, U., Hong, H., and Han, K.-S.: Retrieval pseudo BRDF-adjusted surface reflectance at 440 nm from Geostationary Environmental Monitoring Spectrometer (GEMS), EGUsphere [preprint], https://doi.org/10.5194/egusphere-2024-601, 2024.